# BIFRÖST: 3D-Aware Image compositing with Language Instructions

**Lingxiao Li**[1]    **Kaixiong Gong**[1]    **Weihong Li**[1†]    **Xili Dai**[2]
**Tao Chen**[3]    **Xiaojun Yuan**[4]    **Xiangyu Yue**[1†]

[1]MMLab, The Chinese University of Hong Kong
[2]The Hong Kong University of Science and Technology (Guangzhou)
[3]Fudan University   [4]University of Electronic Science and Technology of China
https://github.com/lingxiao-li/Bifrost

Figure 1: *Bifröst* **results on various personalized image compositing tasks. Top:** *Bifröst* is adept at precise, arbitrary object placement and replacement in a background image with a reference object and a language instruction, and achieves 3D-aware high-fidelity harmonized compositing results; **Bottom Left:** Given a coarse mask, *Bifröst* can change the pose of the object to follow the shape of the mask; **Bottom Right:** Our model adapts the identity of the reference image to the target image without changing the pose.

## Abstract

This paper introduces *Bifröst*, a novel 3D-aware framework that is built upon diffusion models to perform instruction-based image composition. Previous methods concentrate on image compositing at the 2D level, which fall short in handling complex spatial relationships (*e.g.*, occlusion). *Bifröst* addresses these issues by training MLLM as a 2.5D location predictor and integrating depth maps as an extra condition during the generation process to bridge the gap between 2D and 3D, which enhances spatial comprehension and supports sophisticated spatial inter-

---

† Corresponding author

38th Conference on Neural Information Processing Systems (NeurIPS 2024).

actions. Our method begins by fine-tuning MLLM with a custom counterfactual dataset to predict 2.5D object locations in complex backgrounds from language instructions. Then, the image-compositing model is uniquely designed to process multiple types of input features, enabling it to perform high-fidelity image compositions that consider occlusion, depth blur, and image harmonization. Extensive qualitative and quantitative evaluations demonstrate that *Bifröst* significantly outperforms existing methods, providing a robust solution for generating realistically composited images in scenarios demanding intricate spatial understanding. This work not only pushes the boundaries of generative image compositing but also reduces reliance on expensive annotated datasets by effectively utilizing existing resources in innovative ways.

# 1   Introduction

Image generation has flourished alongside the advancement of diffusion models (Song *et al.*, 2021; Ho *et al.*, 2020; Rombach *et al.*, 2022; Ramesh *et al.*, 2022). Recent works (Saharia *et al.*, 2022; Liu *et al.*, 2023b; Brooks *et al.*, 2023; Zhang *et al.*, 2023b; Huang *et al.*, 2023; Li *et al.*, 2023; Chen *et al.*, 2024; He *et al.*, 2024) add conditional controls, *e.g.*, text prompts, scribbles, skeleton maps to the diffusion models, offering significant potentials for controllable image editing. Among these methods for image editing, generative object-level image compositing (Yang *et al.*, 2023; Song *et al.*, 2023; Chen *et al.*, 2024; Song *et al.*, 2024) is a novel yet challenging task that aims to seamlessly inject an outside reference object into a given background image with a specific location, creating a cohesive and realistic image. This ability is significantly required in practical applications including E-commerce, effect-image rendering, poster-making, professional editing, *etc.*

Achieving arbitrary personalized object-level image compositing necessitates a deep understanding of the visual concept inherent to both the identity of the reference object and spatial relations of the background image. To date, this task has not been well addressed. Paint-by-Example (Yang *et al.*, 2023) and Objectstitch (Song *et al.*, 2023) use a target image as the template to edit a specific region of the background image, but they could not generate ID-consistent contents, especially for untrained categories. On the other hand, (Chen *et al.*, 2024; Song *et al.*, 2024) generate objects with ID (identity) preserved in the target scene, but they fall short in processing complicated 3D geometry relations (*e.g.*, the occlusion) as they only consider 2D-level composition. To sum up, previous methods mainly either **1)** fail to achieve both ID preservation and background harmony, or **2)** do not explicitly take into account the geometry behind the background and fail to accurately composite objects and backgrounds in complex spatial relations.

We conclude that *the root cause of aforementioned issues is that image composition is conducted at a 2D level*. Ideally, the composition operation should be done in a 3D space for precise 3D geometry relationships. However, accurately modeling a 3D scene with any given image, especially with only one view, is non-trivial and time-consuming (Liu *et al.*, 2023b). To address these challenges, we introduce *Bifröst*, which offers a 3D-aware framework for image composition without explicit 3D modeling. We achieve this by leveraging depth to indicate the 3D geometry relationship between the object and the background. In detail, our approach leverages a multi-modal large language model (MLLM) as a 2.5D location predictor (*i.e.,* bounding box and depth for the object in the given background image). With the predicted bounding box and depth, our method yields a depth map for the composited image, which is fed into a diffusion model as guidance. This enables our method to achieve good ID preservation and background harmony simultaneously, as it is now aware of the spatial relations between them, and the conflict at the dimension of depth is eliminated. In addition, MLLM enables our method to composite images with text instructions, which enlarges the application scenario of *Bifröst*. *Bifröst* achieves significantly better visual results than the previous method, which in turn validates our conclusion.

We divide the training procedure into two stages. In the first stage, we finetune an MLLM (*e.g.*, LLaVA (Liu *et al.*, 2023a, 2024)) for 2.5D predictions of objects in complex scenes with language instructions. In the second stage, we train the image composition model. To composite images with complicated spatial relationships, we introduce depth maps as conditions for image generation. In addition, we leverage an ID extractor to generate discriminative ID tokens and a frequency-aware detail extractor to extract the high-frequency 2D shape information (detail maps) for ID preservation. We unify the depth maps, ID tokens, and detail maps to guide a pre-trained text-to-image diffusion model to generate desired image composition results. This two-stage training paradigm allows

---

In Norse mythology, a burning rainbow bridge that transports anything between Earth and Asgard.

us to utilize a number of existing 2D datasets for common visual tasks *e.g.*, visual segmentation, and detection, avoiding collecting large-volume text-image data specially designed for arbitrary object-level image compositing.

Our main contributions can be summarized as follows: **1)** We are the first to embed *depth* into the image composition pipeline, which improves the ID preservation and background harmony simultaneously. **2)** We delicately build a counterfactual dataset and fine-tuned MLLM as a powerful tool to predict 2.5D location of the object in a given background image. Further, the fine-tuned MLLM enables our approach to understand language instructions for image composition. **3)** Our approach has demonstrated exceptional performance through comprehensive qualitative assessments and quantitative analyses on image compositing and outperforms other methods, *Bifröst* allows us to generate images with better control of occlusion, depth blur, and image harmonization.

## 2 Related Work

### 2.1 Image compositing with Diffusion Models

Image compositing, an essential technique in image editing, seamlessly integrates a reference object into a background image, aiming for realism and high fidelity. Traditional methods, such as image harmonization (Jiang *et al.*, 2021; Xue *et al.*, 2022; Guerreiro *et al.*, 2023; Ke *et al.*, 2022) and blending (Pérez *et al.*, 2003; Zhang *et al.*, 2020, 2021; Wu *et al.*, 2019) primarily ensure color and lighting consistency but inadequately address geometric discrepancies. The introduction of diffusion models (Ho *et al.*, 2020; Sohl-Dickstein *et al.*, 2015; Song *et al.*, 2021; Rombach *et al.*, 2022) has shifted focus towards comprehensive frameworks that address all facets of image compositing. Methods like (Yang *et al.*, 2023; Song *et al.*, 2023) often use CLIP-based adapters to utilize pretrained models, yet they compromise the object's identity preservation, focusing mainly on high-level semantic representations.More recent studies prioritize maintaining the appearance in generative object compositing. Notable developments in this field include AnyDoor (Chen *et al.*, 2024) and ControlCom (Zhang *et al.*, 2023a). AnyDoor integrates DINOv2 (Oquab *et al.*, 2023) with a high-frequency filter, while ControlCom introduces a local enhancement module, both improving appearance retention. However, these approaches still face challenges in spatial correction capabilities. Most recent work IMPRINT (Song *et al.*, 2024) trains an ID-preserving encoder that enhances the visual consistency of the object while maintaining geometry and color harmonization. However, none of the work can deal with composition with occlusion and more complex spatial relations. In contrast, our models novelly propose a 3D-aware generative model that allows more accuracy and complex image compositing while maintaining geometry and color harmonization.

### 2.2 LLM with Diffusion Models

The open-sourced LlaMA (Touvron *et al.*, 2023; Chiang *et al.*, 2023) substantially enhances vision tasks by leveraging Large Language Models (LLMs). Innovations such as LLaVA and MiniGPT-4 (Liu *et al.*, 2023a; Zhu *et al.*, 2024) have advanced image-text alignment through instruction-tuning. While many MLLM-based studies have demonstrated effectiveness in text-generation tasks like human-robot interaction, complex reasoning, and science question answering, GILL (Koh *et al.*, 2023) acts as a conduit between MLLMs and diffusion models by enabling LLMs to process and generate coherent images from textual inputs. SEED (Ge *et al.*, 2023b) introduces a novel image tokenizer that allows LLMs to handle and concurrently generate images and text, with SEED-2 (Ge *et al.*, 2023a) enhancing this process by better aligning generation embeddings with image embeddings from unCLIP-SD, thus improving the preservation of visual semantics and realistic image reconstruction. Emu (Sun *et al.*, 2024), a multimodal generalist, is trained on a next-token-prediction model. CM3Leon (Yu *et al.*, 2023a), utilizing the CM3 multimodal architecture and adapted training methods from text-only models, excels in both text-to-image and image-to-text generations. Finally, SmartEdit (Huang *et al.*, 2024) proposes a Bidirectional Interaction Module that enables comprehensive bidirectional information interactions between images and the MLLM output, allowing it for complex instruction-based image editing. Nevertheless, these methods requires text-image pairs data to train and do not support accurate spatial location prediction and subject-driven image compositing.

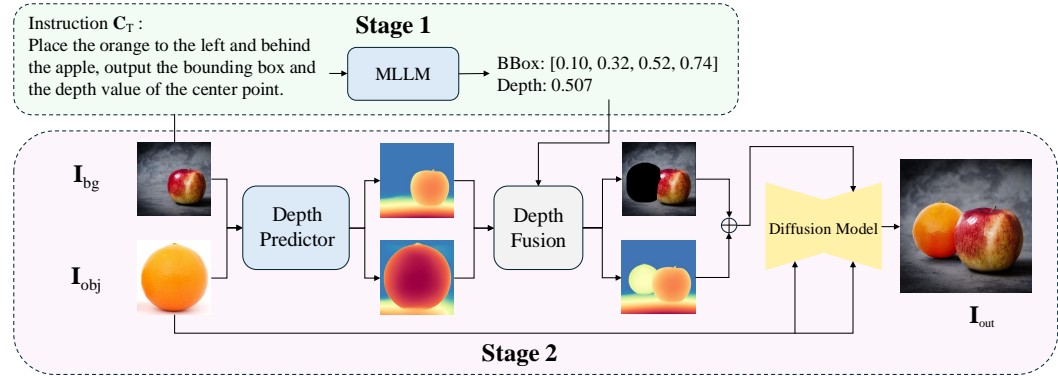

Figure 2: **Overview of the inference pipeline of *Bifröst*.** Given background image $\mathbf{I}_{bg}$, and text instruction $c_T$ that indicates the location for object compositing to the background, the MLLM first predicts the 2.5D location consists of a bounding box and the depth of the object. Then a pre-trained depth predictor is applied to estimate the given images' depth. After that. The depth of the reference object is scaled to the depth value predicted by MLLM and fused in the predicted location of the background depth. Finally, the masked background image, fused depth, and reference object image are used as the input of the compositing model and generate an output image $\mathbf{I}_{out}$ that satisfies spatial relations in the text instruction $c_T$ and appears visually coherent and natural (*e.g.*, with light and shadow that are consistent with the background image).

## 3 Method

The overall pipeline of our *Bifröst* is elaborated in Fig. 2. Our method consists of two stages: 1) in stage 1, given the input image of object and background, and text instruction that indicates the location for object compositing to the background, the MLLMs are finetuned on our customized dataset for predicting a 2.5D location, which provides the bounding box and a depth value of the object in the background; 2) in stage 2, our *Bifröst* performs 3D-aware image compositing according to the generated 2.5D location, images of object and background and their depth maps estimated by a depth predictor. As we divide the pipeline into two stages, we can adopt the existing benchmarks that have been collected for common vision problems and avoid the demand of collected new and task-specific paired data.

We detail our pipeline in the following section. In Sec. 3.1 we discuss building our customized counterfactual dataset and fine-tuning multi-modal large language models to predict 2.5D locations given a background image. Following this, in Sec. 3.2, we introduce the 3D-aware image compositing pipeline that uses the spatial location predicted by MLLM to seamlessly integrate the reference object into the background image. Finally, we discuss combining two stages in Sec. 3.3 and show more application scenarios of our proposed method.

### 3.1 Finetuning MLLM to Predict 2.5D Location

Given a background image $\mathbf{I}_{bg}$ and text instruction $c_T$, which is tokenized as $H_T$, our goal is to obtain the 2.5D coordinate of the reference object we want to place in. We indicate the 2.5D coordinate as $l$ which consists of a 2D bounding box $b = [x_1, y_1, x_2, y_2]$ and an estimated depth value $d \in [0, 1]$. During the training stage, the majority of parameters $\theta$ in the LLM are kept frozen, and we utilize LoRA (Hu *et al.*, 2022) to carry out efficient fine-tuning. Subsequently, for a sequence of length $L$, we minimize the negative log-likelihood of generated text tokens $X_A$, which can be formulated as:

$$L_{\text{LLM}}(\mathbf{I}_{bg}, \boldsymbol{c}_T) = -\sum_{i=1}^{L} \log p_{\{\theta\}}\left(x_i \mid \mathbf{I}_{bg}, \boldsymbol{c}_{T,<i}, X_{A,<i}\right) \tag{1}$$

**Dataset Generation.** However, there is a lack of dataset containing image-text data pairs to teach the MLLM to predict the reasonable location to place in the background image $\mathbf{I}_{bg}$ following the instruction $c_T$ and it is crucial to create such dataset. Since the model needs to predict both a 2D bounding box and depth value in $Z$ axis, this requires the model to be capable of understanding both the spatial relationship between objects and the size relationship between objects (*e.g.*, the bounding box of a dog should be smaller than the car if the user wants to place a dog near the car) and the physical laws in the image (*e.g.*, a bottle should be placed on the table rather than floating in the air).



**Counterfactual Image**  **Masked Image**  **Given Image**  **Predicted Depth**

**Instruction:** Place the laptop to the left of the keyboard,
output the bounding box and the depth value of the center point.

**Answer:**
BBox: [0.00, 0.42, 0.39, 0.98]
Depth: 0.57

Figure 3: **Overview of the 2.5D counterfactual dataset generation for fine-tuning MLLM.** Given a scene image $I$, one object $o$ was randomly selected as the object we want to predict (*e.g.*, the laptop in this figure). The depth of the object is predicted by a pre-trained depth predictor. The selected object is then removed from the given image using the SAM (*i.e.* mask the object) followed by an SD-based inpainting model (*i.e.*, inpaint the masked hole). The final data pair consists of a **text instruction**, a **counterfactual image**, and a 2.5D **location** of the selected object $o$.

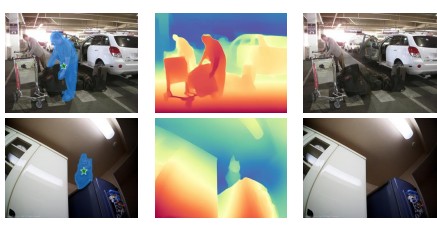

**Instruction**: Place the person in front of the car and to the right of the person, output the bounding box and the depth value of the center point.
**Answer**: [0.32, 0.19, 0.56, 0.87], 0.89.

**Instruction**: Place the cat on top of the refrigerator, output the bounding box and the depth value of the center point.
**Answer**: [0.53, 0.36, 0.65, 0.70], 0.30.

Original Image    Depth Map    **Counterfactual Image**

Figure 4: **Examples of 2.5D counterfactual dataset** for fine-tuning MLLM.

Furthermore, the image should not contain the object we want to place (*i.e.*, the MLLM should learn to predict the location). To this end, we build a novel counterfactual dataset based on MS COCO dataset (Lin *et al.*, 2014). For a background image $x$ and annotation $a$, we use a pre-trained depth predictor DPT (Ranftl *et al.*, 2021) to estimate the depth map $d$. We randomly select $k$ objects and choose one as the target ($o_s$), defining spatial relations with the other $k-1$ objects (*e.g.*, left of, above, in front of). GPT-3 (Brown *et al.*, 2020) generates text descriptions $D$ mentioning all objects and their relative positions, as shown in Fig. 3. The ground truth includes the bounding box and depth value of $o_s$. Using diffusion-based inpainting (Yu *et al.*, 2023b), we remove $o_s$ from $\mathbf{I}_{bg}$. We collect 30,080 image-instruction-answer pairs for training and 855 for testing, with examples in Fig. 4. Further details are in Appendix B.1.

## 3.2  3D-Aware Image compositing

The training pipeline of *Bifröst*'s 3D-aware image compositing module is shown in Fig. 5. Given the reference object, background, and estimated 2.5D location, *Bifröst* extracts ID Token, detail map, and depth maps to generate high-fidelity, diverse object-scene compositions that respect spatial relations. Unlike previous works using only 2D backgrounds and objects, our key contribution is incorporating depth maps to account for spatial relationships. Additionally, we use large-scale data, including videos and images, to train the model to learn the appearance changes, ensuring high fidelity in generated images. Details of different components are as follows.

**ID Extractor.** We leverage pre-trained DINO-V2 (Oquab *et al.*, 2023) to encode images into visual tokens to preserve more information. We use a single linear layer to bridge the embedding space of DINOV2 and the pre-trained text-to-image UNet. The final projected ID tokens can be gotten by:

$$\boldsymbol{c}_i = \mathcal{E}_i \left( \mathbf{I}_{obj} \right), \tag{2}$$

where $\mathcal{E}_i$ indicates the ID extractor.

**Detail Extractor.** The ID tokens inevitably lose the fine details of the reference object due to the information bottleneck. Thus, we need extra guidance for the complementary detail generation. We apply a **high-frequency map** to represent fine-grained details of the object. We further applied **mask shape augmentation** to provide more practical guidance of the pose and view of the generated object, which mimics the casual user brush used in practical editing. The results are shown at the bottom of Fig. 1. After obtaining the high-frequency map, we stitch it onto the scene image at the specified locations and pass the collage to the detail extractor:

$$\boldsymbol{c}_h = \mathcal{E}_h \left( \mathbf{I}_h \right), \tag{3}$$

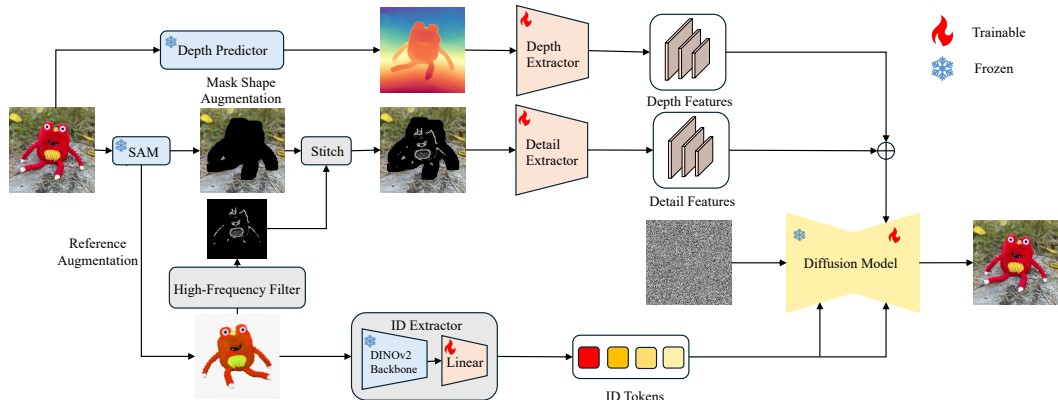

Figure 5: **Overview of training pipeline of *Bifröst* on image compositing stage.** A segmentation module is first adopted to get the masked image and object without background, followed by an ID extractor to obtain its identity information. The high-frequency filter is then applied to extract the detail of the object, stitch the result with the scene at the predicted location, and employ a detail extractor to complement the ID extractor with texture details. We then use a depth predictor to estimate the depth of the image and apply a depth extractor to capture the spatial information of the scene. Finally, the ID tokens, detail maps, and depth maps are integrated into a pre-trained diffusion model, enabling the target object to seamlessly blend with its surroundings while preserving complex spatial relationships.

Table 1: **Statistics of the datasets** used in the image compositing stage.

| | YouTubeVOS | MOSE | VIPSeg | VitonHD | MSRA-10K | DUT | HFlickr | LVIS | SAM (subset) |
|---|---|---|---|---|---|---|---|---|---|
| **Type** | Video | Video | Video | Image | Image | Image | Image | Image | Image |
| **# Samples** | 4453 | 1507 | 3110 | 11647 | 10000 | 15572 | 4833 | 118287 | 178976 |
| **Variation** | ✓ | ✓ | ✓ | ✓ | ✗ | ✗ | ✗ | ✗ | ✗ |

where $\mathcal{E}_h$ is the detail extractor and $c_\mathrm{h}$ is the extracted high-frequency condition. More details can be found in Appendix B.2

**Depth Extractor.** As we mentioned in Sec. 1, all existing image compositing works can only insert objects as foreground. As Fig. 7 shows, the generated images look wired when the inserted object has some overlap with objects in the background. We tackle this problem by proposing a simple yet effective method by adding depth control to the model. By utilizing pre-trained depth predictor DPT (Ranftl *et al.*, 2021), we could generate depth-image pairs without demanding extra annotation. Formally, given a target image $\mathbf{I}_{tar}$, we have

$$c_\mathrm{d} = \mathcal{E}_d \left( \mathrm{DPT}(\mathbf{I}_{tar}) \right), \tag{4}$$

where $\mathcal{E}_d$ is the depth extractor and $c_\mathrm{d}$ is the extracted depth condition. In our experiment, the detail and depth extractors utilize ControlNet-style UNet encoders, generating hierarchical detail maps at multiple resolutions.

**Feature Injection.** Given an image $z_0$, image diffusion algorithms progressively add noise to the image and produce a noisy image $z_t$, where t represents the number of times noise is added. Given a set of conditions including time step $t$, object ID condition $c_i$, high-frequency detail condition $c_h$, as well as a depth condition $c_d$, image diffusion algorithms learn a UNet $\epsilon_\theta$ to predict the noise added to the noisy image $z_t$ with

$$\mathcal{L}_{composite} = \mathbb{E}_{\boldsymbol{z}_0, \boldsymbol{t}, \boldsymbol{c}_i, \boldsymbol{c}_\mathrm{f}, \epsilon \sim \mathcal{N}(0,1)} \left[ \|\epsilon - \epsilon_\theta \left( \boldsymbol{z}_t, \boldsymbol{t}, \boldsymbol{c}_i, \boldsymbol{c}_\mathrm{f} \right) \|_2^2 \right], \tag{5}$$

where $\boldsymbol{c}_f = \boldsymbol{c}_h + \lambda \cdot \boldsymbol{c}_d$ and $\lambda$ is a hyper parameter weight between two controls. Specifically, the ID tokens are injected into each UNet layer via cross-attention. The high-frequency detail and depth maps are first added together and then concatenated with the UNet decoder features at each resolution. During training, the pre-trained UNet encoder parameters are frozen to retain priors, while the decoder is fine-tuned to adapt to the new task.

**Classifier-free Guidance.** To achieve the trade-off between identity preservation and image harmonization, we find that classifier-free sampling strategy (Ho and Salimans, 2022) is a powerful tool.

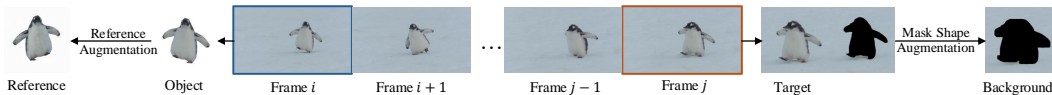

Figure 6: **Data preparation pipeline of leveraging videos.** Given a clip, we first sample two frames, selecting an instance from one frame as the reference object and using the corresponding instance from the other frame as the training supervision.

Table 2: **Quantitative evaluation results** on the accuracy of the MLLM's prediction of *Bifröst*. Note: MiniGPTv2 and LLaVA (baseline) do not support depth prediction.

|  | MiniGPTv2 | LLaVA (baseline) | Ours |
|---|---|---|---|
| **BBox(MSE)** ($\downarrow$) | 0.2694 | 0.0653 | **0.0496** |
| **BBox(IoU)** ($\uparrow$) | 0.0175 | 0.7567 | **0.8515** |
| **Depth (MSE)** ($\downarrow$) | ✗ | ✗ | **0.0658** |

Previous work (Tang *et al.*, 2022) found that the classifier-free guidance is actually the combination of both prior and posterior constraints.

$$\log p\left(\mathbf{y}_t \mid \mathbf{c}\right) + (s-1)\log p\left(\mathbf{c} \mid \mathbf{y}_t\right) \propto \log p\left(\mathbf{y}_t\right) + s\left(\log p\left(\mathbf{y}_t \mid \mathbf{c}\right) - \log p\left(\mathbf{y}_t\right)\right), \quad (6)$$

where $s$ denotes the classifier-free guidance scale. In our experiments, we follow the settings in (Zhang *et al.*, 2023b).

$$\epsilon_{\text{prd}} = \epsilon_{\text{uc}} + s\left(\epsilon_{\text{c}} - \epsilon_{\text{uc}}\right), \quad (7)$$

where $\epsilon_{\text{prd}}$, $\epsilon_{\text{uc}}$, $\epsilon_{\text{c}}$, $s$ are the model's final output, unconditional output, conditional output, and a user-specified weight respectively. In the training process, we randomly replace $50\%$ object ID condition $\mathbf{c}_i$ with empty strings. This approach increases *Bifröst*'s ability to directly recognize semantics in the input conditions as a replacement for the ID tokens. We further replace $30\%$ of the depth map or detail map as blank to increase the robustness of our model. (*i.e.*, our model could generate high-quality images with only one condition, either from detail or depth control).

**Dataset Generation.** The ideal training data consists of image pairs capturing the same object in different scenes and poses, which existing datasets lack. Previous works (Yang *et al.*, 2023; Song *et al.*, 2023) use single images with augmentations like rotation and flip, which are insufficient for realistic pose and view variants. To address this, we use video datasets to capture frames of the same object as complementary following (Chen *et al.*, 2024; Song *et al.*, 2024). As illustrated in Fig. 6, our data preparation pipeline selects two frames from a video and extracts foreground masks. One frame is masked and cropped around the object to serve as the reference, while the other frame—with an augmented mask shape—acts as the background. The unmasked version of this second frame serves as the training ground truth. The dataset quality is another key to better identity preservation and pose variation. The full data used is listed in Tab. 1, which covers a large variety of domains such as nature scenes (SAM (Kirillov *et al.*, 2023), LVIS (Gupta *et al.*, 2019), HFlickr (Cong *et al.*, 2020), DUT (Wang *et al.*, 2017), and MSRA-10K (Borji *et al.*, 2015)), panoptic video segmentation datasets (YoutubeVOS (Xu *et al.*, 2018), VIPSeg (Miao *et al.*, 2022), and MOSE (Ding *et al.*, 2023)), and virtual try-on dataset (VitonHD (Choi *et al.*, 2021)).

### 3.3 Inference

The overall inference pipeline is shown in Fig. 2. Given background image $\mathbf{I}_{bg}$, and text instruction $\mathbf{c}_T$, our fine-tuned MLLM predicts the 2.5D location of object formulated as bounding box $b = [x_1, y_1, x_2, y_2]$ and an estimated depth value $d \in [0, 1]$. Then, we first scale the depth map of the reference image to the predicted depth and fuse it into the bounding box location of the background depth map. The background image is masked following the bounding box. Finally, one can generate the composited image following the pipeline in Fig. 2. We also support various formats of input (*e.g.*, user-specified mask and depth), which allows more application scenarios as Fig. 1 shows. More details can be found in the Appendix B.

## 4 Experiment

### 4.1 Implementation Details

**Hyperparameters.** We choose LLaVA (Liu *et al.*, 2023a, 2024) as our method to fine-tune multi-modal large language models and Vicuna (Chiang *et al.*, 2023) as the LLM. The learning rate is

Table 3: **Quantitive evaluation results** on the performance of image compositing. *Bifröst* outperforms all other methods across all metrics.

| | Paint-by-Example | Object-Stitch | TF-ICON | AnyDoor | Ours |
|---|---|---|---|---|---|
| **DINO-score** ($\uparrow$) | 72.225 | 73.108 | 74.743 | 76.807 | **77.746** |
| **CLIP-score** ($\uparrow$) | 84.584 | 84.836 | 85.627 | 88.735 | **89.722** |
| **FID** ($\downarrow$) | 22.286 | 19.489 | 18.945 | 15.858 | **15.025** |

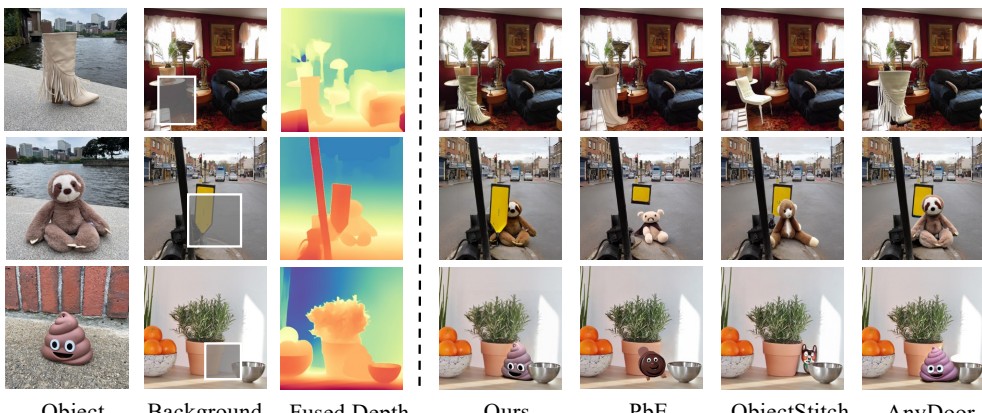

| Object | Background | Fused Depth | Ours | PbE | ObjectStitch | AnyDoor |

Figure 7: **Qualitative comparison with reference-based image generation methods,** including Paint-by-Example (Yang *et al.*, 2023), ObjectStitch (Song *et al.*, 2023), and AnyDoor (Chen *et al.*, 2024), where our *Bifröst* better preserves the geometry consistency. Note that all approaches do not fine-tune the model on the test samples.

set as $2e^{-5}$ and train 15 epochs. We choose Stable Diffusion V2.1 (Rombach *et al.*, 2022) as the base generator for the image compositing model. During training, we set the image resolution to $512 \times 512$. We choose Adam (Kingma and Ba, 2014) optimizer with an initial learning rate of $1e^{-5}$. More details can be found in the Appendix A.

**Zoom-in Strategy.** During inference, given a scene image and a location box, we expand the box into a square using an amplification ratio of 2.0. The square is then cropped and resized to $512 \times 512$ for input into the diffusion model. This approach enables handling scenes with arbitrary aspect ratios and location boxes covering extremely small or large areas.

**Benchmarks.** To evaluate the performance of our Fine-tuned MLLM, we collect 855 image-instruction-answer pairs for testing as we mentioned in Sec. 3.1. For quantitative results of spatial aware image compositing, we follow the settings in (Chen *et al.*, 2024) that contain 30 new concepts from DreamBooth (Ruiz *et al.*, 2023) for the reference images. We manually pick 30 images with boxes in COCO-Val (Lin *et al.*, 2014) for the scene image. Thus, we generate 900 images for the object-scene combinations.

**Evaluation metrics.** We test the IoU and MSE loss to evaluate the accuracy of the predicted bounding box and depth of our MLLM. For the image compositing model, we evaluate performance on our constructed DreamBooth dataset by following DreamBooth (Ruiz *et al.*, 2023) to calculate the CLIP-Score and DINO-Score, which measure the similarity between the generated region and the target object. We also compute the FID (Heusel *et al.*, 2017) to assess realism and compositing quality. Additionally, we conduct user studies with 30 annotators to rate the results based on fidelity, quality, diversity, and 3D awareness.

Table 4: **User study** on the comparison between our *Bifröst* and existing alternatives. "Quality", "Fidelity", "Diversity", and "3D Awareness" measure synthesis quality, object identity preservation, object local variation (*i.e.*, across four proposals), and spatial relation awareness (*i.e.*, occlusion) respectively. Each metric is rated from 1 (worst) to 5 (best).

| | Paint-by-Example | Object-Stitch | TF-ICON | AnyDoor | Ours (*w/o* depth) | Ours (*w/* depth) |
|---|---|---|---|---|---|---|
| **Quality** ($\uparrow$) | 2.24 | 2.66 | 2.75 | 3.57 | 3.64 | **3.96** |
| **Fidelity** ($\uparrow$) | 2.05 | 2.56 | 2.63 | 3.58 | 3.89 | **4.03** |
| **Diversity** ($\uparrow$) | **3.87** | 2.42 | 2.36 | 3.57 | 3.61 | 3.07 |
| **3D Awareness** ($\uparrow$) | 3.56 | 3.37 | 3.42 | 2.51 | 2.56 | **4.21** |

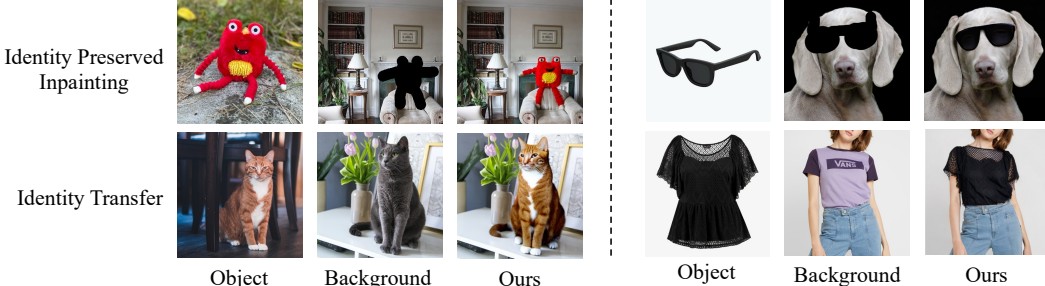

Figure 8: **Results of other application scenarios** of *Bifröst*.

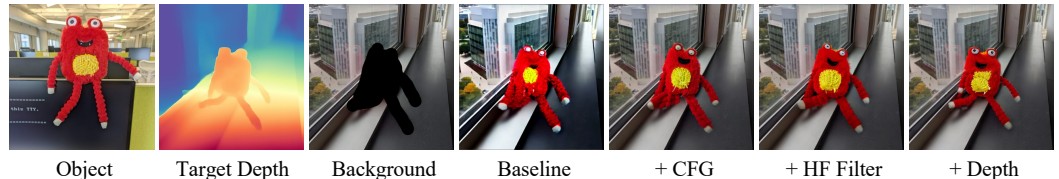

Figure 9: **Qualitative ablation study** on the core components of *Bifröst*, where the last column is the result of our full model, "HF-Filter" stands for the high-frequency filter in the detail extractor.

## 4.2 Quantitative Evaluation

**MLLM Evaluation.** To evaluate the effectiveness of our fine-tuned MLLM, we compare our model with two vison-language LLMs: LLaVA (Liu *et al.*, 2023a) and miniGPTv2 (Zhu *et al.*, 2024). However, to our knowledge, none of the MLLM support accurate depth prediction. The results are shown in Tab. 2, which indicates that our fine-tuned MLLM outperforms other models significantly on the spatial bounding box prediction task.

**Spatial-Aware Image compositing Evaluation.** To demonstrate the effectiveness of our model, we test our model and four existing methods (Paint-by-Example (Yang *et al.*, 2023), ObjectStitch (Song *et al.*, 2023), TF-ICON (Lu *et al.*, 2023)), and AnyDoor (Chen *et al.*, 2024) on the dreambooth (Ruiz *et al.*, 2023) test sets. The same inputs (a mask and a reference object) are used in all models. As shown in Tab. 3, *Bifröst* consistently outperforms baselines across all metrics, indicating that our model generates images with both realism and fidelity.

## 4.3 Qualitative Evaluation

To better evaluate the performance of our spatial-aware image compositing model, we qualitatively compare our method against prior methods as shown in Fig. 7. PbE and ObjectStitch show natural compositing effects, but they fail to preserve the object's ID when the object has a complex texture or structure. Although AnyDoor maintains a fine-grained texture of the object, it can not handle occlusion with other objects in the scene. In contrast, our model achieves better ID preservation and demonstrates flexibility in adapting to the background even in complex scenes with occlusion(*i.e.*, the poop emoji in the third row is seamlessly injected between the bowl and flowerpot with no artifact). We also provide more results of other application scenarios in Fig. 8.

**User Study.** We organize user study to compare Paint-by-Example, ObjectStitch, TF-ICON, AnyDoor, and our model. The user-study results are listed in Tab. 4. It shows that our model owns evident superiorities for fidelity, quantity, and 3D awareness, especially for 3D awareness. Without depth control, our model can generate more diverse objects adjusted for the background. Nevertheless, the quality fidelity, and 3D awareness degrade significantly if the depth control is removed. However, as (Yang *et al.*, 2023) only keeps the semantic consistency but our methods preserve the instance identity, they naturally have larger space for the diversity. In this case, *Bifröst* still gets higher rates than (Song *et al.*, 2023; Lu *et al.*, 2023) and competitive results with (Chen *et al.*, 2024), which verifies the effectiveness of our method. This results indicate that introducing the depth information is the key for *Bifröst* to achieve both high fidelity and 3D-aware image compositing. More details are in the Appendix F.

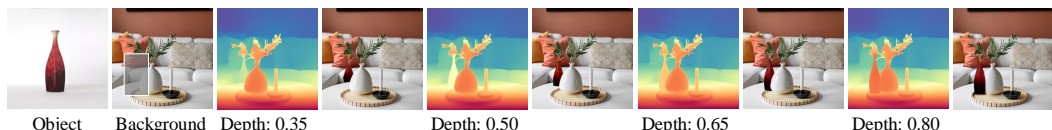

| Object | Background | Depth: 0.35 | Depth: 0.50 | Depth: 0.65 | Depth: 0.80 |

Figure 10: **Ablation study** of different depth control from deep to shallow.

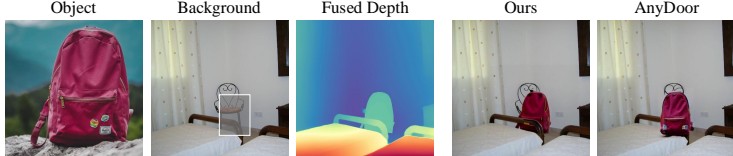

| Object | Background | Fused Depth | Ours | AnyDoor |

Instruction: Place the backpack behind the bed.

Figure 11: **Failure case** from the out-of-distribution dataset and comparison with AnyDoor.

## 4.4 Ablation Study

Tab. 5 shows the effect of components in *Bifröst*. Performance drops significantly without video data during training, as it cannot adjust poses and views to adapt to unseen scenes with only image data. Fig. 9 visualizes results of removing different components. CFG achieves a trade-off between identity preservation and image harmonization, significantly boosting performance. Setting the collage region from the high-frequency map to an all-zero map evaluates the HF Filter's contribution, showing it effectively guides the generation of fine structural details. Adding depth control further improves the fidelity and quality of generated images, as visualized in Fig. 10, where the red vase moves from behind the table to the front of the white vase as depth increases.

Table 5: **Quantitative ablation studies** on the core components of the image compositing model of *Bifröst*. Note that: + indicates adding one component based on the previous model.

|  | Baseline | +Video Data | +CFG | +HF Filter | +Depth |
|---|---|---|---|---|---|
| **DINO-score** ($\uparrow$) | 68.693 | 71.573 | 75.578 | 76.372 | **77.746** |
| **CLIP-score** ($\uparrow$) | 80.458 | 83.536 | 86.394 | 88.634 | **89.722** |
| **FID** ($\downarrow$) | 20.465 | 17.168 | 15.837 | 15.342 | **15.025** |

## 4.5 Limitations and Future Work

Our *Bifröst* achieves high-quality, instruction-based image compositing but has limitations. **1)** While our fine-tuned MLLM performs well on in-domain test datasets, it struggles with OOD datasets containing untrained objects and more complex scenes. A failure case is illustrated in Fig. 11, where the model intends to position the backpack behind the bed. While MLLM predicts the correct location, it overlaps with the chair in the background. Nonetheless, our approach still outperforms prior work (Chen *et al.*, 2024), which fails to handle occlusions between the bed and chair due to insufficient understanding of spatial and depth relationships. This issue can be addressed by increasing the size of the training dataset (*e.g.*, utilizing large-scale, OpenImages (Kuznetsova *et al.*, 2018). **2)** The use of depth maps significantly enhances our model's generation capabilities, enabling precise control over object placement and complex spatial relationships. However, this reliance on depth maps restricts the diversity of generated objects, particularly in terms of novel poses and views. Although we employ classifier-free guidance during inference to provide some flexibility, it represents a trade-off between spatial control and object diversity. This can potentially be solved by robust depth control during the training stage. We leave this for future work.

## 5 Conclusion

In conclusion, *Bifröst* represents a significant advancement in object-level image compositing. Leveraging the capabilities of powerful MLLM, our framework successfully addresses the limitations of existing SD-based image compositing methods, facilitating their transition from research to practical applications. Our experimental results demonstrate that *Bifröst* not only achieves state-of-the-art quality and fidelity in instruction-based, object-level image compositing but also provides a controllable generation process that accommodates complex spatial relationships. Future research endeavors encompass the refinement of the MLLM through the utilization of more extensive datasets and extending our framework to support 3D and video-based personalized object compositing tasks, further broadening its applicability and enhancing its performance.

**Acknowledgements** This work is partially supported by the National Natural Science Foundation of China (Grant No. 62306261), CUHK Direct Grants (Grant No. 4055190), and The Shun Hing Institute of Advanced Engineering (SHIAE) No. 8115074. We thank Boqing Gong (Google) for feedback on an early draft of this paper and Yuanyuan Zhang (University of Liverpool) for proposing the name *Bifröst* of our model.

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

# Supplementary Material

## Overview

This appendix is organized as follows:

Appendix A gives more implementation details of *Bifröst*. Sec 4.1

Appendix B provides more technical details and mathematical formulae used in Sec 3.1 & Sec 3.2

Appendix C explains more details of our evaluation details in experiments. Sec 4.1

Appendix D provides more details of creating counterfactual dataset and gives more examples from the constructed dataset. Sec 3.1

Appendix E shows more visual results and comparison with prior methods. Sec 4.3

Appendix F shows more details of the user study we conducted. Sec 4.3

Appendix G discusses the ethic problems might be caused by *Bifröst* and potential positive societal impacts and negative societal impacts.

## A   Implementation Details

For fine-tuning MLLM, following the base setting in (Liu *et al.*, 2023a), we choose Vicuna (Chiang *et al.*, 2023) as our LLM. We use pre-trained CLIP-ViT-large (Radford *et al.*, 2021) as the visual encoder. The learning rate is set at $4e^{-5}$ to avoid over-fitting. The batch size is set as 16, we train 15 epochs with 4 A800 GPUs, which takes about 5 hours to finish the fine-tuning.

For 3D-aware image compositing model. We use Stable Diffusion 2.1 as the pre-trained diffusion model. The model can also be initialized from the pre-trained weights of (Chen *et al.*, 2024) to better utilize the prior knowledge. We choose Adam (Kingma and Ba, 2014) optimizer with an initial learning rate of $1e^{-5}$. The DINO-V2 (Oquab *et al.*, 2023) ID extractor and the encoder of the U-net are frozen during the training. The decoder of the U-net, detail extractor, and depth extractor are fine-tuned during this process. The batch size is set as 16, we train 20k steps on 4 A800 GPUs, which takes 1 day to finish the training.

Although training our model requires considerable computing resources as mentioned in Appendix Section A, the runtime cost and resources required for the inference stage are affordable. Our model can run on a single NVIDIA RTX 4090 GPU (24GB) thanks to our two-stage training/inference since one does not need to load all models simultaneously. The total inference time for one image composition can be finished in 30 seconds on an RTX 4090 (these include predicting the depth map, running the SAM model to remove the background of the reference image, using MLLM to predict the 2.5D location, depth fusion, and image composition, the DDIM sampler is set as 50 steps).

## B   Method

### B.1   MLLM Fine Tuning

Given a background image $\mathbf{I}_{bg}$ and text instruction $c_T$, which is tokenized as $H_T$, our goal is to obtain the 2.5D coordinate of the reference object we want to place in, indicates $l$ consists of a 2D bounding box $b = [x_1, y_1, x_2, y_2]$ and an estimated depth value $d \in [0, 1]$. following the base setting in (Liu *et al.*, 2023a), we choose Vicuna (Chiang *et al.*, 2023) as our LLM $f_\theta(\cdot)$ parameterized by $\theta$, As shown in Fig. 3. For an input background image $\mathbf{I}_{bg}$, we first use the pre-trained visual encoder to encode the image, which provides the visual feature $Z_V = g(\mathbf{I}_{bg})$. Then, we use a simple linear layer to connect image features into the word embedding space. Specifically, we apply a trainable projection matrix $W$ to convert $Z_V$ into language embedding tokens $H_V$, which have the same dimensionality as the word embedding space in the language model:

$$H_V = W \cdot Z_V, \text{with } Z_V = g(\mathbf{I}_{bg}) \tag{8}$$

Thus, we have a sequence of visual tokens $H_V$, which will be sent to the LLM with text tokens $H_T$.

**Counterfactual Dataset Generation** In practice, we choose the depth value of the central points of the selected object bounding box as the target depth value for MLLM to predict. The reason we do

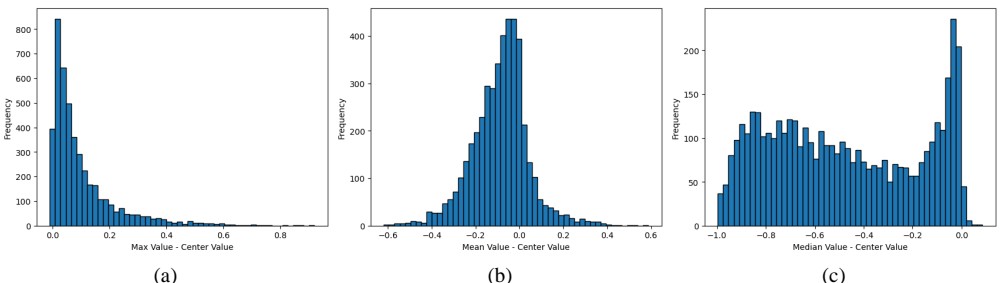

(a)            (b)            (c)

Figure 12: **The distribution of differences between center value and three other choices of the depth values,** where (a) is the differences between the max value of depth and center depth value; (b) is the differences between the mean value of depth and center depth value; and (c) is the differences between the median value of depth and center depth value

not use the entire depth value is that the goal of predicting the depth value of the reference object is to estimate the location (in depth dimension) of the object in the background image for the final image composition. To this end, it does not have to be very accurate as long as it is in a reasonable range and a single depth value for the object is shown to be sufficient. Though it is possible to estimate the depth value for the entire object (e.g., via DPT), it is more costly and challenging to construct the dataset (pixel-wise depth annotation) and train the MLLM to predict the depth value for the entire object. We will explore more sufficient strategies for this in the future. However, there are other choices for points to represent the 2.5D location of the selected object besides the center point. There are various options for estimating the depth value, such as the depth value at the center point of the bbox, the maximum, average, and median depth value of the object within the bbox, etc. They all have their pros and cons and result in similar performance. For instance, the median value varies significantly along with the size of the object in the bounding box. The average value may be influenced by extreme values in the bounding box. Furthermore, the maximum value might also be influenced by the extreme value in the bounding box. We conducted an extra experiment in Fig. 12 to compare the differences between different choices. We calculate the differences between different choices of the depth value we want to predict on 5000 examples in the evaluation dataset. The results show that the difference between the maximum depth value and the center point value is small. Most of the differences are less than 0.2. While the differences between the average value and the center point value basically follow a Gaussian distribution with a mean value of -0.05. However, the median value is not reliable compared with other choices. In this work, as we focus on general settings where reference objects are from common categories, we use the depth value of the bbox center point, which works well in our experiments. Optimizing the location for depth value estimation may be helpful for objects from specific categories, which are rare, and we leave this in future work.

## B.2 3D-Aware Image Compositing

**ID Extractor**

We employ pre-trained visual encoders to extract the target object's identity. Rather than using CLIP (Radford *et al.*, 2021) for object embedding (Yang *et al.*, 2023; Song *et al.*, 2023), prior work (Chen *et al.*, 2024; Song *et al.*, 2024) demonstrates that DINO-V2 (Oquab *et al.*, 2023) better captures discriminative features, projecting objects into an augmentation-invariant feature space. Thus, we use DINO-V2 to encode the image into a global token $\mathbf{T}g^{1\times1536}$ and patch tokens $\mathbf{T}p^{256\times1536}$. These tokens are concatenated to retain more information. Following (Chen *et al.*, 2024), a linear layer bridges the DINO-V2 and pre-trained text-to-image UNet embedding spaces, yielding final ID tokens $\mathbf{T}_{\text{ID}}^{257\times1024}$.

**Collage Representation.** Unlike prior methods (Casanova *et al.*, 2023; Sarukkai *et al.*, 2024; Chen *et al.*, 2024) that use full-color images or high-frequency maps as complementary information, we found that these approaches often produce objects that resemble the reference image too closely, creating a "copy-paste" effect. To address this, we remove color information from the HF-map in (Chen *et al.*, 2024), enhancing fine-grained details and improving the visual coherence of the generated objects.

**Mask Shape Augmentation.** To provide greater user control, we define five levels of coarse masks, including a bounding box mask, as shown in Fig. 14. As the mask level increases (from 1 to 5), the model gains more flexibility in object generation.

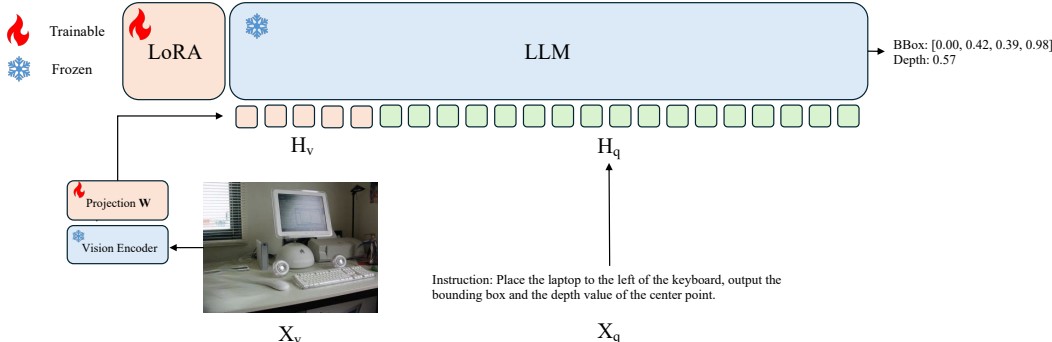

Figure 13: **Overview of the MLLM fine-tuning.**

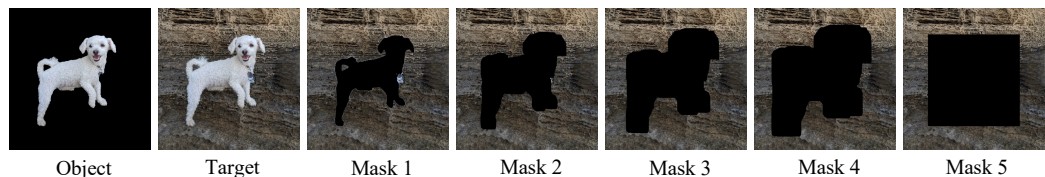

Figure 14: **Different types of mask used in the image compositing stage.** The generation is constrained in the masked area, so the user-provided mask is able to modify the pose, view, and shape of the subject.

The **Detail Extract** process is formulated as:

$$\mathbf{I}_h = (\mathbf{I}_{obj} \otimes \mathbf{K}_h + \mathbf{I}_{obj} \otimes \mathbf{K}_v) \odot \mathbf{M}_{\text{aug}}, \qquad (9)$$

where $\mathbf{K}_h$ and $\mathbf{K}_v$ are horizontal and vertical Sobel kernels (Kanopoulos *et al.*, 1988) serving as high-pass filters. Here, $\otimes$ and $\odot$ represent convolution and Hadamard product, respectively. Given an object image $\mathbf{I}_{obj}$, high-frequency regions are extracted through these filters, and a shape-augmented mask $\mathbf{M}_{\text{aug}}$ is applied to remove information near the outer contour of the object.

## B.3 Inference

As discussed in Sec. 3, *Bifröst* supports multiple types of inference modes. We illustrate different inference modes in Fig. 15.

**ID Transfer.** To transfer the identity of an object in the background, we first employ a segmentor to accurately mask the target object while maintaining the original depth of the background image. The model then generates a new image using the reference object, background depth map, and masked background.

**ID Preserved Inpainting.** To alter the pose or view of the reference object, we keep the background image's depth unchanged and use the reference object, background depth map, and masked background as inputs to generate a new image.

**Place.** To place an object into the background, we use the fine-tuned MLLM to predict the precise bounding box and depth value for the desired location. The reference object's depth map is scaled accordingly and fused into the background depth map at the specified location, with the mask shape adjusted to reflect this fusion, as shown in the third row of Fig. 15. The model then generates a new image using the reference object, background depth map, and masked background.

**Replace.** To replace an object in the background with a reference object, we first scale the reference object's depth map to match the depth value predicted by the MLLM and fuse the depth maps. The entire bounding box is masked, setting the value to 0 outside the object to ensure it is fully masked. The background image is also fully masked within the bounding box. The model then generates a new image using the reference object, background depth map, and masked background.

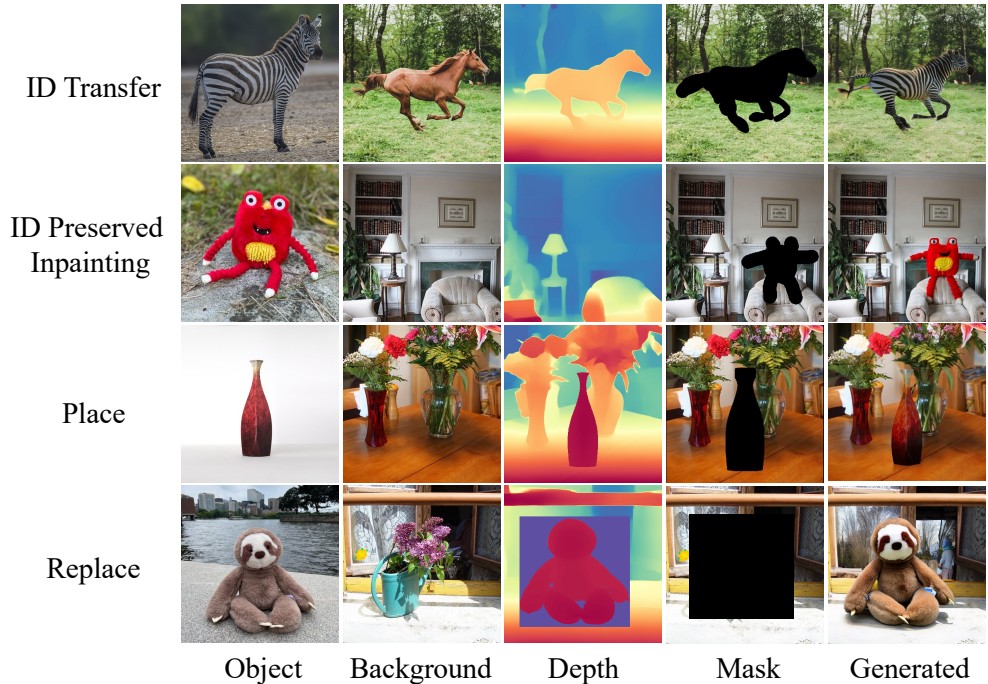

| Object | Background | Depth | Mask | Generated |
|--------|-----------|-------|------|-----------|

Figure 15: **Comparison of different inference modes**

## C Evaluation Details

To fairly compare the quality of the generated images between different methods, we cropped the generated images to the given bounding box area and compared the similarity with the reference objects.

## D Counterfactual Dataset

In this section, we provide more details on building the counterfactual dataset using a COCO (Lin *et al.*, 2014) dataset. We take Fig. 16 as an example. For a given image with multiple bounding boxes. We randomly select an object as the object we want to predict (*i.e.*, bowl with grape in the red bounding box). Then we randomly select multiple objects as the objects we want to describe relative location with (*i.e.*, carrot and spoon with green bounding boxes in this picture). Then we define the relative relations between the selected object and other objects based on their spatial relation. In Fig. 16, the grape bowl is to the left of the spoon and underneath the carrot. The relation definition process also considers the depth information; if the depth values are different between two objects, we will define the relations as "behind" or "in front of" based on the real situation. Then we prompt GPT3 (Brown *et al.*, 2020) to generate the instruction text. However, the usage of GPT is not mandatory. The usage of GPT3 is only for sample text instructions given spatial relations and object names, which will not use the "reasoning" ability of GPT. This simple task can even be done by a naive approach by setting pre-defined instruction templates and filling the blanket with spatial relations and object names. The ground truth answer consists of the bounding box of the selected object and the depth value for the center of the object. Finally, we remove the selected object as the right image in Fig. 16 shows. We show more examples of our counterfactual dataset in Fig. 17.

## E More Qualitative Results

We show more qualitative comparison with Paint-by-Example (Yang *et al.*, 2023), ObjectStitch (Song *et al.*, 2023), and AnyDoor (Chen *et al.*, 2024) in Fig. 18. PbE and ObjectStitch show natural compositing effects, but they fail to preserve the object's ID when the object as lines 2-4 show. Although AnyDoor maintains a fine-grained texture of the object, it can not handle occlusion

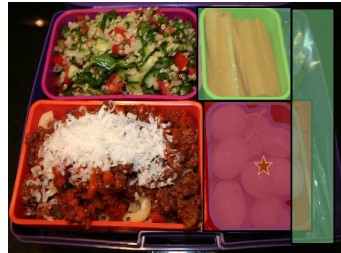 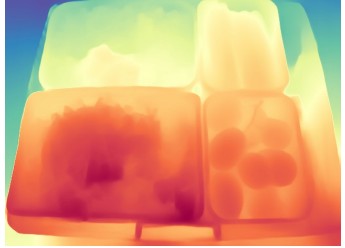 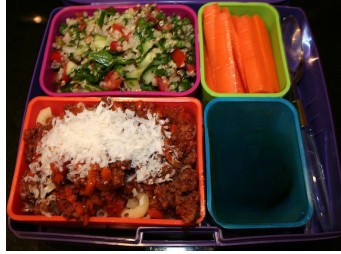

**Instruction**: Place the bowl to the left of the spoon and underneath the carrot, output the bounding box and the depth value of the center point.
**Answer**: [0.60, 0.38, 0.93, 0.91], 0.78.

Figure 16: **More examples of 2.5D counterfactual dataset building process**

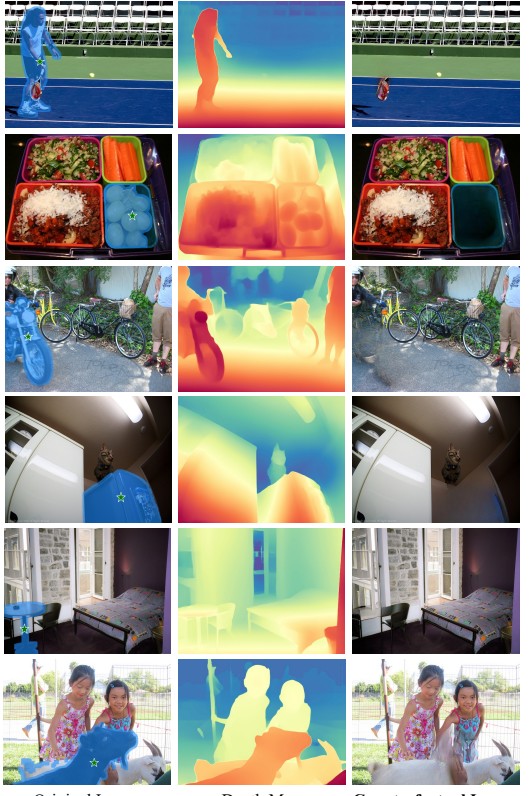

**Instruction**: Place the person in front of the chair, output the bounding box and the depth value of the center point.
**Answer**: [0.09, 0.06, 0.33, 0.90], 0.80.

**Instruction**: Place the bowl to the left of the spoon and underneath the carrot, output the bounding box and the depth value of the center point.
**Answer**: [0.60, 0.38, 0.93, 0.91], 0.78.

**Instruction**: Place the motorcycle in front of the person, output the bounding box and the depth value of the center point.
**Answer**: [0.00, 0.18, 0.27, 0.94], 0.94.

**Instruction**: Place the refrigerator under the cat, output the bounding box and the depth value of the center point.
**Answer**: [0.47, 0.60, 0.92, 0.99], 0.73.

**Instruction**: Place the dining table to the left of the bed, output the bounding box and the depth value of the center point.
**Answer**: [0.00, 0.60, 0.24, 1.00], 0.46.

**Instruction**: Place the sheep in front of the person, output the bounding box and the depth value of the center point.
**Answer**: [0.08, 0.64, 1.00, 1.00], 0.71.

Original Image     Depth Map     **Counterfactual Image**

Figure 17: **More examples of 2.5D counterfactual dataset** for Fine-tuning MLLM.

with other objects in the scene. In contrast, our model perfectly handles occlusion in complex environments(*i.e.*, the dogs in lines 1 and 4 behind the hydrant).

We further provide more examples of images generated by *Bifröst* given text instructions. The results show that it seamlessly injects the object into the backgrounds while satisfying the instructions. For instance, the lights and shallows change with the environment. It also perfectly handles the occlusion with other objects.

Following the same setup in prior works (Ruiz *et al.*, 2023; Chen *et al.*, 2024; Song *et al.*, 2024), the training and testing data are exclusive for image composition. This has evaluated the OOD ability of the image composition of our method, indicating the good generalization ability of our method. As mentioned in the limitation section in our paper, the OOD ability for the MLLM in stage 1 can be affected by the number of object and scene categories in the dataset in stage 1 for predicting 2.5D location (the estimated depth may not be very accurate but still in a reasonable range for OOD objects

or scenes). However, as we only require a rough depth value for image composition, the effects of the OOD issue for image composition in stage 2 are relatively minor.

To further verify this, we report the results of OOD objects and scenes in Fig. 19, though both objects and scenes have not been seen in stage 1 (e.g., piano and church are not included in MS-COCO), the MLLM can still predict reasonable depth values for image composition.

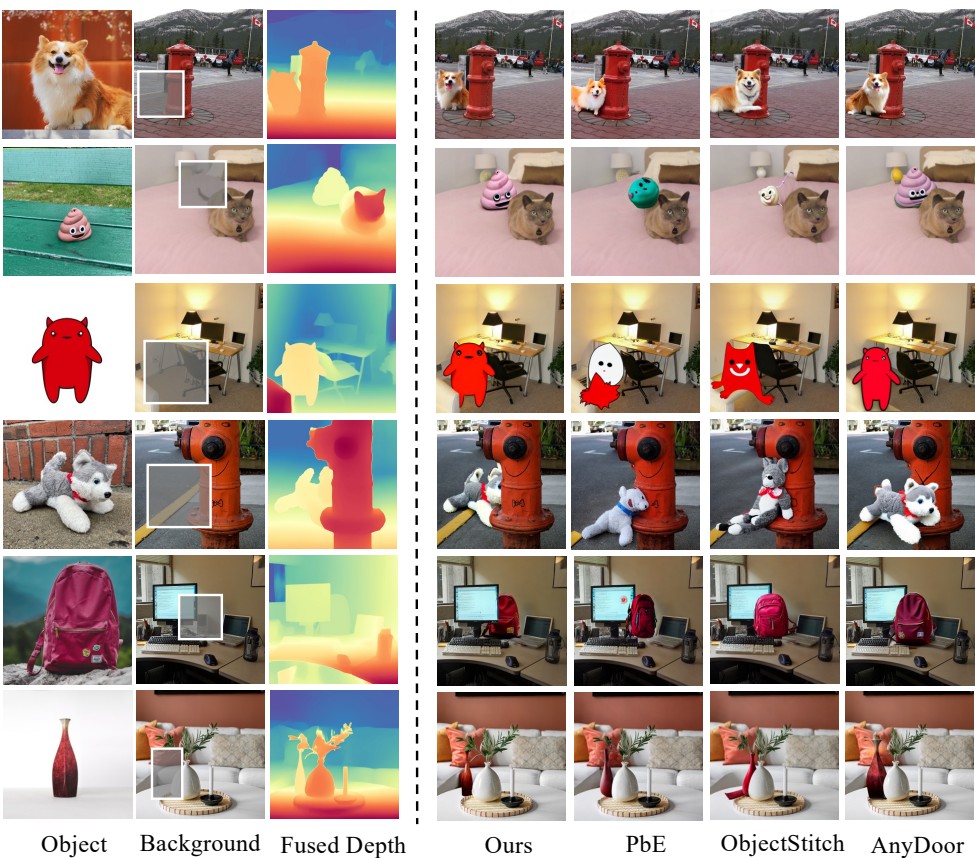

|  Object | Background | Fused Depth | Ours | PbE | ObjectStitch | AnyDoor |

Figure 18: **More qualitative comparison with reference-based image generation methods,** including Paint-by-Example (Yang *et al.*, 2023), ObjectStitch (Song *et al.*, 2023), and AnyDoor (Chen *et al.*, 2024).

## F  Human Evaluation Interface

As mentioned, we conducted an extensive user study with a fully randomized survey. Results are shown in the main text. Specifically, we compared *Bifröst* with four other models Paint-by-Example (Yang *et al.*, 2023), ObjectStitch (Song *et al.*, 2023), TF-ICON (Lu *et al.*, 2023), Any-Door (Chen *et al.*, 2024):

1. We chose 30 images from DreamBooth (Ruiz *et al.*, 2023) test set, and for each image, we then generated 3 variants with different backgrounds using each model, respectively. Overall, there were 30 original images and 90 generated variants in total.

2. For each sample of each model, we present one masked background image, a reference object, and the generated image to annotators. We then shuffled the orders for all images.

3. We recruited 30 volunteers from diverse backgrounds and provided detailed guidelines and templates for evaluation. Annotators rated the images on a scale of 1 to 5 across four criteria: "Fidelity", "Quality", "Diversity", and "3D Awareness". "Fidelity" evaluates identity preservation, while "Quality" assesses visual harmony, independent of fidelity. "Diversity" measures variation among generated proposals to discourage "copy-paste" style

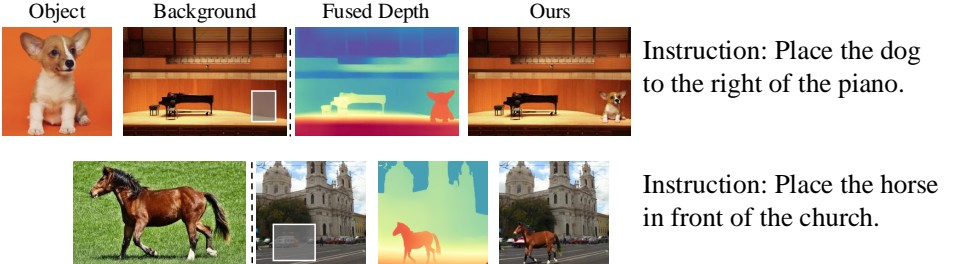

Instruction: Place the dog to the right of the piano.

Instruction: Place the horse in front of the church.

Figure 19: **Examples that both object and background are from the out-of-distribution dataset.**

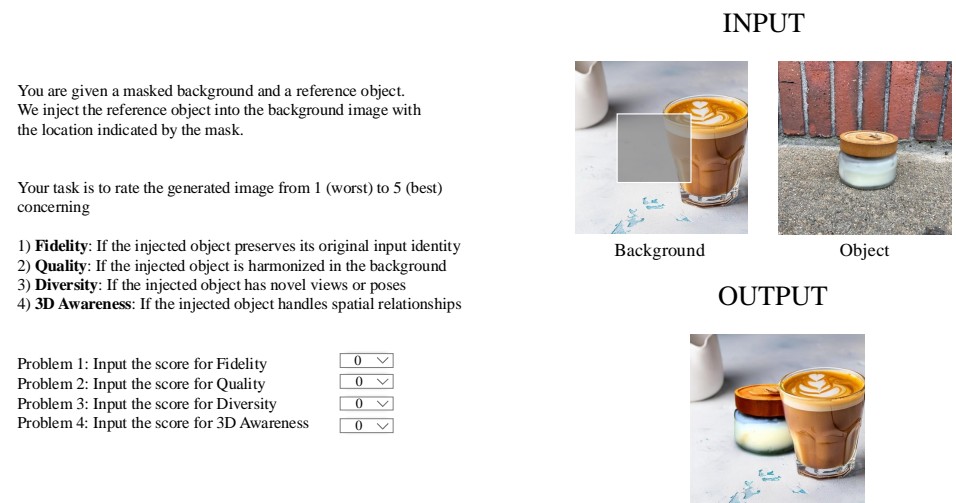

Figure 20: **The illustration of the user study interface.**

outputs. "3D Awareness" evaluates the object's ability to handle spatial relationships (*e.g.*, occlusion) seamlessly.

The user-study interface is shown in Fig. 20.

## G   Ethics Discussion

The advancements in image generation (Ruiz *et al.*, 2023; Rombach *et al.*, 2022) and compositing (Chen *et al.*, 2024; Song *et al.*, 2023, 2024) through our proposed method offer significant positive societal impacts. Our approach enhances practical applications in fields such as e-commerce, professional editing, and digital art creation. By enabling precise and realistic image compositing, our method can improve user experience, facilitate creative expression, and streamline workflows in various industries. Additionally, the ability to use text instructions for image compositing makes technology more accessible to non-experts, promoting inclusivity in digital content creation. However, there are potential negative societal impacts to consider. The misuse of realistic image compositing technology could lead to the creation of deceptive or harmful content, such as deepfakes or misleading advertisements. This raises ethical concerns regarding privacy, consent, and the potential for misinformation. To mitigate these risks, it is crucial to establish guidelines and ethical standards for the use of such technology. Furthermore, transparency in the development and deployment of these models is necessary to build trust and ensure responsible usage. In summary, while our method presents valuable advancements in image compositing, careful consideration and proactive measures are essential to address the ethical implications and prevent potential misuse.

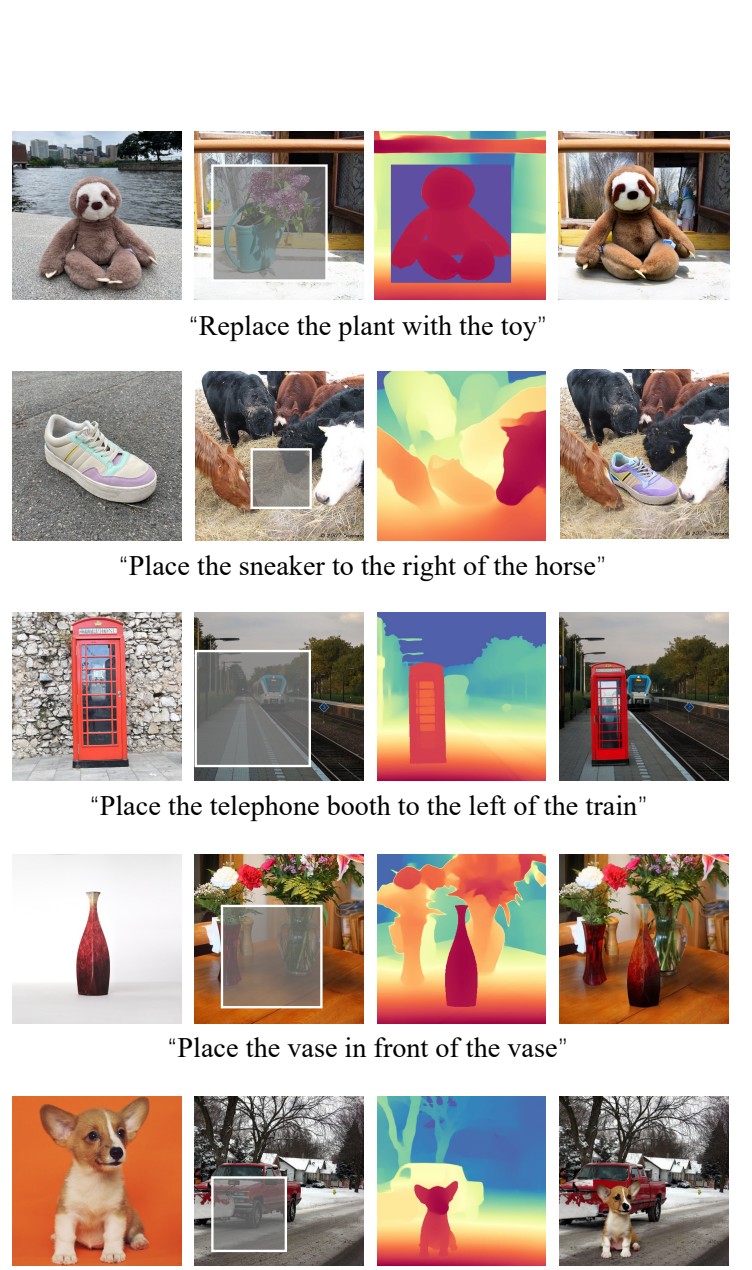

"Replace the plant with the toy"

"Place the sneaker to the right of the horse"

"Place the telephone booth to the left of the train"

"Place the vase in front of the vase"

"Place the dog in front of the truck"

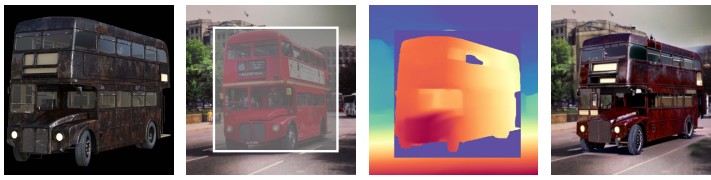

"Replace the bus with the bus"

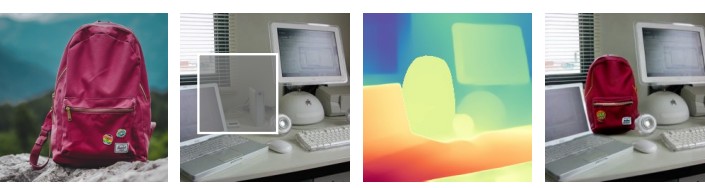

"Place the backpack to the right of the laptop"

Figure 21: **More results generated by *Bifröst* with language instructions**

