# OpenReview forum: "$\textit{Bifr\"ost}$: 3D-Aware Image Compositing with Language Instructions"
_NeurIPS.cc/2024/Conference — NeurIPS 2024 poster_

### Official Review · Reviewer_dLCn · 2024-07-12

**Soundness:** 4
**Presentation:** 4
**Contribution:** 4
**Rating:** 7
**Confidence:** 4

**Summary:**

This paper proposes a 3D-aware framework for generative image compositing. It first fine-tune a MLLM with custom counterfacual dataset consisting of image-instruction-answer triplets to predict object bounding box and depth value. In the second stage, a pretrained diffusion model is fine-tuned with large-scale custom image. Experiments show that the proposed framework presents a great spatial understanding ability.

**Strengths:**

- The practical design of leveraging MLLM (for 2.5D location prediction) and pretrained diffusion models for 3D-aware object compositing is of great impact.
- The dataset generation process for finetuning of both MLLM and diffusion models is inspirational.
- The paper is well written and carefully presented.
- Experiments are solid and extensive.

**Weaknesses:**

- How was the image-instruction-answer triplet data generated? Line 150-151 didn't describe the instruction generation process clearly. Also description of how instructions are generated in figure 3 caption is ambiguous.
- Typo in figure 3 caption: "Fine-tuning"->"fine-tuning"
- Font in figure 5 could be larger.

**Questions:**

Please see weakness.

**Limitations:**

The authos have adequately addressed the limitations and societal impact.

---

> ### Author Rebuttal · Authors · 2024-08-07
>
> # Response to Reviewer dLCn
>
> We thank the reviewer for the positive feedback and insightful comments. We address the questions and comments below.
>
> > Q1: ...image-instruction-answer triplet data generation…
>
> As we described in the Dataset Generation section. We have gotten the names of the selected objects (“laptop” in the example in Figure 3 in the main paper), the names of the objects that have spatial relations with (“keyboard” in the example), and the spatial relations with these objects (“to the left of” in the example). Then, we ask GPT3 to generate a text instruction that includes these key components. An example of the final output can be “Place the laptop to the left of the keyboard, output the bounding box and the depth value of the center point.” However, the usage of GPT is not mandatory. The usage of GPT3 is only for sample text instructions given spatial relations and object names, which will not use the “reasoning” ability of GPT. This simple task can even be done by a naive approach by setting pre-defined instruction templates and filling the blanket with spatial relations and object names. Therefore, this is totally reproducible and scalable. We will release the script to reproduce the dataset.
>
> > Q2: typos
>
> Thanks, we will proofread the paper and correct the errors in the revised version.
>
> > Q3: font size
>
> Thanks for the suggestion, it would be better if the font size was larger. We will enlarge the font size in the final version.

---

> > ### Comment · Reviewer_dLCn · 2024-08-12
> >
> > I have read all the other reviewers' comments and the authors' responses. My concerns have been addressed by the authors. I will not reduce my score.

---

> > > ### Author Response · Authors · 2024-08-13
> > >
> > > Dear Reviewer dLCn,
> > >
> > > Thanks for your reply! We appreciate your positive comments and will incorporate your suggestions in the revised version.
> > >
> > > Best wishes,
> > >
> > > Authors

---

### Official Review · Reviewer_CdZU · 2024-07-13

**Soundness:** 3
**Presentation:** 1
**Contribution:** 4
**Rating:** 6
**Confidence:** 4

**Summary:**

This paper presents a 3D-aware image compositing framework to guide the depth-aware placement of objects within images. The method leverages a multi-modal LM which proposes a bounding box and depth of the object to be inserted. This is followed by fusion reference object depth with the background's depth map. Then a diffusion model conditioned on the mask and fused depth map produces the final output. The method aims to address limitations in existing 2D compositing methods by enhancing 3D spatial comprehension. The paper claims significant improvements in usability, visual quality, and realism over previous methods.

**Strengths:**

* Innovation. The integration of depth maps and the use of a multi-modal large language model (MLLM) for 2.5D location predictions are innovative. These elements are well-justified and represent a significant advancement over existing 2D compositing methods. This basically solves the occlusion issues in previous methods.
* Qualitative results. Results shown in the paper demonstrate clear improvements in the realism and quality of the composed images. I particularly like the study on varying depth values (Fig. 10).

**Weaknesses:**

* Clarity. The paper writing is poor and feels rushed. There are several typos and uncommon word choices that distract the reader. Some aspects are inadequately described (particularly depth fusion during inference) and are not entirely clear.
* Quantitative results. The authors show metrics/user study for quality, fidelity etc but do not present any quantitative results on '3D awareness', which is the main goal of the paper. For example: Since fused depth maps are available, authors can compare object-level depth error compared to previous works like AnyDoor.
* Generalization: Generalization is an important desired quality of compositing methods. The paper mentions limitations with out-of-domain datasets but does not provide sufficient details or analysis on how significant this issue is. How does the generalization compare with related works? More empirical evidence and discussion on generalization would strengthen the paper.
* Computational Efficiency: There is limited discussion on the computational cost of the proposed method. Resources required for inference. Runtime?

Minor comments / Typos:

"Image composing" is a rather uncommon phrase. A more common term is "image compositing" or "image composition".
"Depth extractor" can be confused as a model used to predict depth. "Depth feature extractor" or "Depth encoder" can be more appropriate. Similar for "detail extractor".

Fig 5 also contains confusing naming. (Depth map -> depth features?) (detail map -> detail features?)

L133: "...object we want to place in, indicates l consists of a 2D bounding box..." This sentence doesn't make sense grammatically.

L180: "cd = Ed (DPT(Itar)), where Ed is the depth extractor and ch is the extracted depth condition." ch -> cd

L226: "During training, we process the image resolution to 512 × 512." "process" -> "set"

L245: "However, to our knowledge, none of the MLLM support accuracy depth prediction." "accuracy" ->"accurate"?


L215: "We first scale the depth map of the reference image to the predicted depth" How?
Depth fusion is vague and inadequately described.

User study is a quantitative result and should come under Sec 4.2

Fig 7 row 1 col 2. Bbox doesn’t match with predictions made by the method.

**Questions:**

* How does the generalization compare with related works?
* How does the method compare with related works quantitatively in terms of 3D awareness?
* what is runtime cost and resources required for inference?
* What is the depth value used for the box? center or average value? Center can miss the object completely? (e.g. circular or objects with holes). How is this situation avoided?

**Limitations:**

yes

---

> ### Author Rebuttal · Authors · 2024-08-07
>
> # Response to Reviewer CdZU
>
>
> We thank the reviewer for the positive feedback and insightful comments. We address the questions and comments below.
>
> > Q1: ...clarity…
>
> R1: Thanks, we will proofread the paper and correct the writing errors. Regarding our depth fusion algorithm, we briefly introduce it, and details are provided in supplementary B3 (our code is attached) due to the space limitation in the main paper. Here, we provide the pseudo-code for depth fusion:
>
>      depth fusion(ref_depth, bg_depth, BBox, depth_range):
>          # def_depth: the depth map of the reference object
>          # bg_depth: the depth map of the background image
>          # BBox: bounding box predicted by the MLLM in format (x, y, w, h)
>          # depth_range: [B-A, B] where A is a hyperparameter set by users, the default value is 0
>          #  		      B is the target depth predicted by the MLLM
> 	     # resize the depth map of the reference object to the shape of the bounding box
> 	      resized_ref_depth = reshape((bbox[2], bbox[3]), ref_depth)
> 	      scaled_ref_depth = scale(max=B, min=B-A, resized_ref_depth)
> 	      for i in range(w):
> 		     for j in range(h):
> 			     if scaled_ref_depth[i, j] > bg_depth[x+i, y+j]:
> 				     bg_depth[x+i, y+j] = scaled_ref_depth[i, j]
> 			     end if
> 		     end for
> 	     end for
> 	     return bg_depth
>
>
> > Q2: ...quantitative results…3D-awareness…
>
> R2: Prior works for image composition methods are not aware of the geometry relations between the reference objects and scenes (or objects in the scene) and fails to composite images when the relations are complicated to them (e.g., Figure 7 in the main paper, when inject an image behind an object of the background image, existing methods suffer from object occlusions). Unlike them, our method is able to place the object in the correct location in spatial and depth dimensions (Figure 7), which indicates the 3D-awareness of our method. And the object level depth errors of previous works will be much higher than ours, i.e., prior works place the reference images in the wrong location (The last column In Figure 7 in the main paper shows that the prior work, AnyDoor, generated the final image where the object is not behind the bowl as required, leading to higher depth errors than ours).  We can see that some alternative methods including Paint-by-Example (PbE) and ObjectStitch can also handle the occlusion issue to some extent (but they fail to generate the correct reference objects) and we conduct user study for comparisons of quality, fidelity, diversity. We further conduct user study for evaluating the 3D-awareness reported in Table 6 in  the attached pdf in 'Response to All Reviewers', which indicates the 3D-awareness of our method.
>
> > Q3: ...generalization ability…
>
> R3: Following the same setup in prior works[A,B,C], the training and testing data are exclusive for image composition. This has evaluated the OOD ability of the image composition of our method, indicating the good generalization ability of our method. As mentioned in the limitation section in our paper, the OOD ability for the MLLM in our stage 1 can be affected by the number of object and scene categories in the dataset in stage 1 for predicting 2.5D location (the estimated depth may not be very accurate but still in a reasonable range for OOD objects or scenes). However, as we only require a rough depth value for image composition, the effects of OOD issue for image composition in stage 2 are relatively minor.
>
> To further verify this, we report the results of OOD objects and scenes in the attached pdf in the 'Response to All Reviewers'. As we can see in Figure 20, though both objects and scenes have not been seen in stage 1 (e.g., piano and church are not included in MS-COCO), the MLLM can still predict reasonable depth values for image composition. However, in some cases, like the example in Figure 21 where the method should put the backpack behind the bed, though the location predicted by MLLM is behind the bed, the location overlaps with the chair in the background. However, our method still obtains better performance than prior work [B] in this case, and they fail to deal with the occlusion with the bed and chair in the image due to the lack of understanding of the relations of objects and scenes in spatial and depth dimensions.  We will add this discussion to more clearly clarify the generalization ability of our method in the final version.
>
> > Q4: ...computational effciency…
>
> R4: Please kindly refer to the answer of Q4 in the 'Response to All Reviewers'.
>
>
> > Q5: ...depth value used for the box…
>
> R5: We use the center value for prediction. Please refer to the answer of Q3 in the 'Response to All Reviewers'. Such rare cases will not influence the evaluation performance of our MLLM. The objects in the MS COCO dataset are common. Thus, there is no (or very small number of) such rare cases in the training and evaluation dataset we created. Therefore, there is no such kind of noisy data that degrades the performance of our MLLM. The MLLM learns to predict reasonable 2.5D locations from the training data without seeing the reference image we want to place. As a result, during the inference stage, the MLLM will still predict correct locations that satisfy the spatial relations in the text instructions without considering the shape of the input objects.
>
>
> [A] DreamBooth: Fine Tuning Text-to-Image Diffusion Models for Subject-Driven Generation, CVPR, 2023
>
> [B] AnyDoor: Zero-shot Object-level Image Customization, in CVPR, 2024
>
> [C] IMPRINT: Generative Object Compositing by Learning Identity-Preserving Representation, in CVPR, 2024

---

> > ### Comment · Reviewer_CdZU · 2024-08-12
> >
> > I appreciate the authors providing the user study results on '3D awareness' and clarifying other issues. The authors have acknowledged the writing errors and promised to improve them in the final version. Most of my concerns have been addressed, and I recommend accepting this paper.

---

> > > ### Author Response · Authors · 2024-08-13
> > >
> > > Dear Reviewer CdZU,
> > >
> > > Thanks for raising your score and for the support of our work and it's contributions! We greatly appreciate the time you spent reviewing our paper, and going through the rebuttal and the other reviews, as well as your valuable suggestions for improving our work. We will correct the writing errors and incorporate extra experiments in the revised version.
> > >
> > > Best wishes,
> > >
> > > Authors

---

### Official Review · Reviewer_wo4W · 2024-07-22

**Soundness:** 3
**Presentation:** 3
**Contribution:** 2
**Rating:** 6
**Confidence:** 4

**Summary:**

This paper deals with a “generative fill” task where the generation model aims to fill the reference object to the target location of a background image in a reasonable flavor. The first difficulty is to figure out the exact target location and the relative depth for the reference object to appear in the generated image. The authors dealt with this problem by fine-tuning a MLLM on a collected counterfactual dataset. The second difficulty is to preserve both general and detailed information of the reference object and condition the diffusion model to produce a plausible result. The proposed method disentangles various visual elements by using existing depth predictor and segmentation networks and recompose those visual representations obtained to a unified generated image through a diffusion model.

**Strengths:**

1. The authors provided access to their codes and dataset, which can be beneficial to the community, if open-sourced afterwards.

2. The proposed model establishes a good result and positions higher than compared methods.

**Weaknesses:**

1. The proposed method lacks technical novelty. I’m not sure whether this paper is the first to embed depth into the image composition pipeline but it is a common practice in the community to incorporate depth, such as ControlNet, to nominate a popular one. Classifier-free guidance and using videos as training data are seemingly popular techniques and their efficacy is widely proved. Fine-tuning MLLM using LoRA is also common. Therefore, the major techniques are not new, nor offering new knowledge to the community.

2. The proposed method has a decent reliance on various large models, which means that the inference runtime and deployment to restricted devices can be an issue.

3. The method assumes the reference object to have a single depth value across all pixels. This assumption may not work for large objects like trains in a multi-layer image.

**Questions:**

1. For the MLLM evaluation, how do LLaVa and miniGPTv2 compare to the proposed model? Are they tuned or what? If not tuned, it is almost for certain that they are not comparable to a tuned one (the proposed).

2. The paper mentioned that they chose DINOv2 as an ID extractor since other papers used the same thing. But did the authors try other models? This is a curiosity question and is not related to the final consideration of rating so the authors should feel comfortable not to answer this question if they feel space is a limitation.

**Limitations:**

The paper proposed a dataset and built the model by using their crafted dataset. It might make sense to check whether the way and the source to build the dataset could lead to data privacy and copy right issues.

---

> ### Author Rebuttal · Authors · 2024-08-07
>
> # Response to Reviewer wo4W
>
> We thank the reviewer for the positive feedback and insightful comments. We address the questions and comments below.
>
> > Q1: ...lacks technical novelty…
>
> R1: Many thanks for acknowledging that our method obtains good results than other models and for the contribution of our code and datasets. In this work, we show that our method achieves controllable image generation by incorporating depth as a condition, while prior image composition or text-driven image editing works suffer from object occlusion problems, e.g., existing works fail when injecting an image behind an object in the background images. To the best of our knowledge, we are the first to embed depth information for image composition. We believe that our work and results can inspire future works to consider depth as a valuable source of prior information for image composition. Additionally, the counterfactual dataset collected in this work also enables the MLLM model to predict the 2.5D location of the reference object in a given background image and will also be beneficial beyond this task, such as 3D localization and robotics (navigation, grasping, etc).
>
> > Q2: …inference runtime…
>
> R2: Although training our model requires considerable computing resources as mentioned in Appendix Section A, the runtime cost and resources required for the inference stage are affordable. Our model can run on a single NVIDIA RTX 4090 GPU (24GB) thanks to our two-stage training/inference since one does not need to load all models simultaneously. The total inference time for one image composition can be finished in 20 seconds on an RTX 4090 (these include predicting the depth map, running the SAM model to remove the background of the reference image, using MLLM to predict the 2.5D location, depth fusion, and image composition, the DDIM sampler is set as 50 steps). We will release our code and pre-trained models to the public.
>
> > Q3: …a single depth value across all pixels…
>
> R3: In our method, we provide a hyperparameter to control the range of the depth of the reference objects, and it is set as 0 by default (i.e., all pixels are set to the predicted depth). And we can see that this works well for most cases. In case of compositing large objects like trains in a multi-layer image, one can adjust the hyperparameter (by cross-validation) to a larger value to allow a wider depth range for large objects. The details are provided in our submitted code (line 390 in run_inference.py) and we will clarify this in the final version.
>
> > Q4: …how do LLaVa and miniGPTv2 compare to the proposed model…
>
> R4: LLaVA and MiniGPT2 are not fine-tuned as they are able to predict a location (bounding box) given the text instruction, especially LLaVA. And we show the comparisons between them and our fine-tuned MLLM for demonstrating the effectiveness of our created dataset for understanding the relations of objects in a scene and improving this ability of existing models.
>
> > Q5: …why DINOv2…
>
> R5: We use DINOv2 as the ID extractor as in the prior works (e.g., AnyDoor and IMPRINT [A, B] ) where the effects of various image encoders such as CLIP, DINOv2, and MLP for ID extractors have been investigated and the analysis shows the superiority of DINOv2. We will better clarify this in the final version.
>
> [A] AnyDoor: Zero-shot Object-level Image Customization, in CVPR, 2024
>
> [B] IMPRINT: Generative Object Compositing by Learning Identity-Preserving Representation, in CVPR, 2024

---

> > ### Comment · Reviewer_wo4W · 2024-08-12
> > **Official comment by Reviewer wo4W**
> >
> > Thanks for addressing my concerns. Most of the original concerns are well addressed. For the single depth value issue, I think it might make sense to show the experiments of hyperparameter adjustment as proposed by the authors in the revision. I will raise my score to accept.

---

> > > ### Author Response · Authors · 2024-08-13
> > >
> > > Dear Reviewer wo4W,
> > >
> > > Thank you for raising your score and for supporting our work and its contributions! We sincerely appreciate the time you dedicated to reviewing our paper, considering the rebuttal, and analyzing the other reviews. We will incorporate additional experiments of hyperparameter adjustment in the revised version.
> > >
> > > Best wishes,
> > >
> > > Authors

---

### Official Review · Reviewer_M1TS · 2024-07-22

**Soundness:** 3
**Presentation:** 3
**Contribution:** 2
**Rating:** 5
**Confidence:** 4

**Summary:**

## Summary of the Paper:

*Problem Statement*:

     Given a single-object image I_ref with object O_ref , a background image I_bg containing an object O_bg, and a text prompt P_text, the paper proposes a method for composing O_ref onto I_bg in such a way that the position of O_ref in the resulting image adheres to the text query P_text.


*Motivation*:

     Existing works on image composition, which mainly relied on processing images in the 2D domain, underperformed in the presence of object occlusion (O_ref in I_ref) . This paper, named BiFrost, argues that image composition algorithms can benefit from processing beyond the 2D domain. A natural choice would be leveraging priors from the 3D domain.  However, due to the complexity of single-image-to-3D analysis, the paper propose to use 2.5D information by incorporating depth information into their pipeline.

*Contributions*:

1)	 Use of depth information in image composing task.
2)	 A new dataset of image-text pairs for object location prediction task i.e.to predict 2D bbox and depth value of the object.
3)      Leveraging multi-modal large language models (MLLM) into their pipelines


*Dataset used*:

1)	 For predicting 2.5D information: Newly created dataset of image-text pairs based on MS-COCO dataset
2)	 For image composing task: 9 datasets -- 3 video-based, 6 image-based datasets. (this setting is similar to previous work in this area)


*Underlying Modeling Tool*:

     Transformer is the common modeling tool among all involved tasks, including depth map generation (GPT3, DPT from CVPR 2021, SAM and Diffusion-based Inpainting Model), MLLM fine-tuning (using LLaVa), and image composition task (DPT, SAM, DINO_V2 and Stable Diffusion)



*Learning Mechanism*:

     Strongly supervised


*Loss Functions*:

     For fine-tuning MLLM, negative log-likelihood of the generated text tokens

     For training image composing task, standard diffusion loss i.e. L2 loss between ground truth noise added to image at time step t and predicted noise at that step.

Quantitative Metric:

     DINO-score, CLIP-score, FID, User study


*Baselines*:

     On the image composition task, four existing methods are compared against: Paint-by-Example, ObjectStitch, TF-ICON, AnyDoor.

**Strengths:**

1)	 Well-written paper
2)	 Incorporating depth information to account for occlusion in text-driven image composition task is an interesting idea
3)	 A new dataset for predicting bounding box and depth values for a given image and a text prompt  -- such dataset, if made public, will be useful to the research community working on image composing task

**Weaknesses:**

1)	 It seems that the paper is trying to force-fit large foundation models for a simple task of incorporating depth information. What was the reasoning behind this? What other alternatives were explored?
2)	 Almost all of the components are based on large foundation models and have frozen weights. It leaves the paper with a single technical contribution, i.e., the use of 2.5D information, instead of 2D.
3)	 The proposed pipeline involves fine-tuning a MLLM, which is in-turn, used to predict only a single depth value. Since SAM is used to get object masks, why not use depth values on the entire object?
4)	 Two-stage training -- I was wondering why stage-1 (where the MLLM is finetuned) cannot be merged with stage-2 to obtain a single end-to-end training pipeline? The reasoning being that since MLLM is tasked to predict only the bounding box and a single depth value
5)	 Using a single depth value can be erroneous. For e.g., in Figure 3, the star mark on “masked image” shows that the center point of laptop is on the corner of the laptop. There could be cases where the center point is NOT on the object of interest. How does the model perform in such scenarios?
6)    Due to the many sub-modules involved in the method, starting from training a depth predictor module using GPT3, I suspect the method is not easily extendable, nor replicable within a certain margin error. I want to understand the authors' stand here.

**Questions:**

1)     How do you calculate center point for predicting depth value? In my understanding it is the center of the bbox. Is that correct?
2)	 Why just a single depth point, why not multiple points or object mask? As mentioned earlier, a single point can be erroneous.
3)	 Is it assumed that the reference object image I_ref will always have one object? Are there any images during train/test time with multiple objects?
4)	 When creating the dataset for MLLM finetuning, why use MS-COCO dataset? Why not use a subset of the training data from image composing task itself? How does training on MS-COCO dataset help in predicting better depth values, for, say, YouTubeVOS or HFlickr dataset?
5)	 Is the model trained on all nine datasets combined OR separately,  and test only on 900 object-scene combination images based on DreamBooth and COCO-Val set? Again, my question is why different datasets for training and test? Are they from same distribution?

**Limitations:**

Yes, limitations of the work have been discussed.

No societal impact can be seen from the work.

---

> ### Author Rebuttal · Authors · 2024-08-07
>
> # Response to Reviewer 1M1TS
>
> We thank the reviewer for the positive feedback and insightful comments. We address the questions and comments below.
>
> > Q1: …incorporating depth…reasoning behind this…
>
> R1: Many thanks for acknowledging our novelty in the context of image composition. We achieve 3D-aware image composition by incorporating depth information, which provides the location information in the depth dimension to enable 3D-aware image composition, e.g., avoiding a crash in image composition due to the occlusion issue. We have explored alternative strategies for this problem. We have attempted to generate scenes layer by layer and place the reference object at the target layer according to the language instructions.  However, this does not well solve the problem as decomposing scenes into layers itself is a challenging research problem. Alternatively, one can embed the depth information predicted by [A] directly into the generative models. It fails due to the lack of ability to understand the relations between the reference object and the target background. We will include this in the final version.
>
>
> > Q2: …single technical contribution…
>
> R2: Please refer to the answer of Q1 in the 'Response to All Reviewers'.
>
> > Q3: …depth…entire object…
>
> R3: The goal of predicting the depth value of the reference object is to estimate a location (in depth dimension) of the object in the background image for the final image composition. To this end, it does not have to be very accurate as long as it is in a reasonable range and a single depth value for the object is shown to be sufficient. Though it is possible to estimate the depth map for the entire object (e.g., via DPT), it is more costly and challenging to construct the dataset (pixel-wise depth annotation) and train the MLLM to predict the depth map for the entire object. We will explore more efficient strategies for this in the future.
>
>
> > Q4: …end-to-end training..
>
> R4: Yes, it's possible to merge stage 1 with stage 2. However, merging both stages will inevitably increase training costs and difficulty due to the lack of high-quality paired instruction-original image-compositing image data. Collecting such a dataset for end-to-end training for both stages is complicated and extremely expensive. We formulate our method into 2 stages, making it easier and cheaper for collecting or using existing data for training our method for each stage, as explained in the paper (line 63-66, line 122-124).
>
> > Q5: …center point is NOT on the object of interest…
>
> R5: Please refer to the answer of Q3 in the 'Response to All Reviewers'. In this work, as we focus on general settings where reference objects are from common categories, we use the depth value of the bbox center point, which works well in our experiments. Optimizing the location for depth value estimation may be helpful for objects from specific categories which are rare, and we leave this in future work. We will add this discussion in our final version.
>
>
> > Q6: …using GPT3…not easily extendable…
>
> R6: GPT3 is only used for sampling text instructions given spatial relations and object names for collecting the new dataset at stage 1. And this does not use the “reasoning” ability of GPT. This step can be replaced by a simple approach by setting pre-defined instruction templates and filling the blanket with spatial relations and object names or alternatives. To this end, this is reproducible and scalable. We will release the dataset and scripts for reproducing it.
>
> > Q7: …reference object…have one object? Are there…multiple objects?
>
> R7: We follow exactly the same setting in prior work [C,D], where the reference object only has one object, and the setting of multiple objects is not considered in this work. The method can be simply extended to multiple reference objects by cropping objects and compositing objects into the background image one by one. We leave this in our future work.
>
> > Q8: …why use MS-COCO dataset…How does…YouTubeVOS or HFlickr?
>
> R8: Stage 1 aims to predict 2.5D information indicating the location of the reference object in the background image. This requires the MLLM model in stage 1 to be able to understand different relations (spatial and depth dimensions) between scenes and objects. To this end, we use MS-COCO, which covers diverse common scenes and object categories in daily life. We do not use the dataset used in the image composition in stage 2 because it contains fewer object categories and fewer object-scene/object relations, e.g., many images do not have objects or only one object.
>
> We thank the reviewer for suggesting two more large datasets that are primarily used for video segmentation. We find that MS-COCO may be better than YTBVOS and HRlickr for stage 1 as MS-COCO covers more object, scene, and relation types. However, it is possible that merging several datasets can further benefit the tasks, and we leave this in the future work.
>
> > Q9: Is the model trained on all nine datasets combined OR separately…
>
> R9: The model was trained on all nine datasets combined, and the training and testing datasets are from different data distributions as the prior works (e.g., AnyDoor, IMPRINT,). We follow exactly the same settings in prior works and use DreamBooth and COCO val as the testing datasets for fair comparisons with previous methods. This setting aims to evaluate image customization or composition methods’ ability in out-of-distribution scenarios [B,C,D]. We will better clarify this in the final version.
>
> [A] Harnessing the Spatial-Temporal Attention of Diffusion Models for High-Fidelity Text-to-Image Synthesis, in CVPR, 2023.
>
> [B] DreamBooth: Fine Tuning Text-to-Image Diffusion Models for Subject-Driven Generation, CVPR, 2023
>
> [C] AnyDoor: Zero-shot Object-level Image Customization, in CVPR, 2024
>
> [D] IMPRINT: Generative Object Compositing by Learning Identity-Preserving Representation, in CVPR, 2024

---

> > ### Author Response · Authors · 2024-08-13
> >
> > # Comment
> >
> > Dear Reviewer 1M1TS,
> >
> > Thanks again for your time in reviewing our work.
> >
> > This is a gentle reminder that the Discussion period will end on 13 Aug 11:59 pm AoE. There is only one day remaining. Would you mind checking our responses to your concerns and questions and confirming whether you have any other questions? We are glad to answer any further questions you may have before the deadline.
> >
> > Best regards,
> >
> > Authors

---

> > > ### Comment · Reviewer_M1TS · 2024-08-14
> > >
> > > Authors,
> > >
> > > Thanks for the reminder.
> > >
> > > I have read through the rebuttal. I see most of my questions have been answered.
> > >
> > > However, for Q5, I do not agree that the midpoint of bbox always lies on the object of interest for common categories. Thing of an office desk with no drawers. The response to Q5 is not convincing, and infact, factually incorrect.
> > >
> > > As well, similar questions regarding the viability of predicting a single depth value were raised by other reviewers.
> > >
> > > As such, I am not inclined on increasing the score of the paper. It will be. BA from my side.

---

> > > > ### Author Response · Authors · 2024-08-14
> > > >
> > > > Dear Reviewer 1M1TS,
> > > >
> > > > We express gratitude for your time spent reviewing our work and your valuable comments.
> > > >
> > > > In fact, as we discussed in the common response, although the midpoint does not always work, it works most of the time since we focus on general settings where reference objects are from common categories. $\textbf{The rare cases will not influence the performance of the fine-tuned MLLM}$ for predicting reasonable locations that satisfy spatial relations in the language instructions.
> > > >
> > > > There are various options for estimating the depth value, such as the depth value at the center point of the bbox, the maximum, average, and median depth value of the object within the bbox, etc. $\textbf{They all have their pros and cons and result in similar performance}$. For instance, the median value varies significantly along with the size of the object in the bounding box. The average value may be influenced by extreme values in the bounding box. Furthermore, the maximum value might also be influenced by the extreme value in the bounding box. We conducted an extra experiment in Figure 19 in the attached PDF file to compare the differences between different choices. We calculate the differences between different choices of the depth value we want to predict on 5000 examples in the evaluation dataset.
> > > >
> > > > Therefore, $\textbf{we can not find a perfect choice that works well in all cases}$. Optimizing the location for depth value estimation may be helpful for objects from specific categories which are rare, and we leave this in future work. We will add this discussion in our final version.
> > > >
> > > > We will be happy to answer your question if you have any further concerns about our work!
> > > >
> > > >
> > > > Best regards,
> > > >
> > > > Authors

---

### Author Rebuttal · Authors · 2024-08-07

# Response to all Reviewers

We thank the reviewers for their valuable time and feedback, and for acknowledging the well writing (Reviewer M1TS, dLCn), interesting ideas (Reviewer M1TS), solid results (Reviewer M1TS, CdZU, dLCn), and novel contribution (Reviewer M1TS, CdZU, dLCn). To our best knowledge, our work is the first attempt to embed depth into image composition and enables 3D-aware compositing images (e.g., our method can understand the spatial and depth relations and generate images with both realism and fidelity). We address the questions and comments below and we will make our code and datasets publicly available for the final version.

> Q1: Novelty and Technical Contributions (To Reviewer M1TS, wo4W)

R1: Many thanks for acknowledging the novelty in the context of image composition. In this work, we show that our method achieves controllable image generation by incorporating depth as condition, while prior image composition or text-driven image editing works suffer from object occlusion problems e.g., existing works fail when injecting an image behind an object in the background images. Additionally, the counterfactual dataset collected in this work also enables the MLLM model to predict the 2.5D location of the reference object in a given background image and will also be beneficial beyond this task, such as 3D localization and robotics (navigation, grasping etc).

> Q2: …predict only a single depth value but not the entire object…  (To Reviewer M1TS, CdZU)

R2: The goal of predicting the depth value of the reference object is to estimate the location (in depth dimension) of the object in the background image for the final image composition. To this end, it does not have to be very accurate as long as it is in a reasonable range and a single depth value for the object is shown to be sufficient. Though it is possible to estimate the depth value for the entire object (e.g., via DPT), it is more costly and challenging to construct the dataset (pixel-wise depth annotation) and train the MLLM to predict the depth value for the entire object. We will explore more efficient strategies for this in the future.

> Q3: Why the center point of the bounding box but not other points?  (To Reviewer M1TS, CdZU)

R3: As mentioned in Q2 above, the predicted depth value indicates the location in depth dimension of the reference object in the background image and it does not have to be very precise as long as it is in a roughly reasonable range and a single depth value is sufficient. There are various options for estimating the depth value, such as the depth value at the center point of the bbox, the maximum, average, and median depth value of the object within the bbox, etc. They all have their pros and cons and result in similar performance. For instance, the median value varies significantly along with the size of the object in the bounding box. The average value may be influenced by extreme values in the bounding box. Furthermore, the maximum value might also be influenced by the extreme value in the bounding box. We conducted an extra experiment in Figure 19 in the attached PDF file to compare the differences between different choices. We calculate the differences between different choices of the depth value we want to predict on 5000 examples in the evaluation dataset. The results show that the difference between the maximum depth value and the center point value is small. Most of the differences are less than 0.2. While the differences between the average value and the center point value basically follow a Gaussian distribution with a mean value of -0.05. However, the median value is not reliable compared with other choices. In this work, as we focus on general settings where reference objects are from common categories, we use the depth value of the bbox center point, which works well in our experiments. Optimizing the location for depth value estimation may be helpful for objects from specific categories which are rare, and we leave this in future work. We will add this discussion in our final version.


> Q4: Runtime Cost and resources required for inference  (To Reviewer wo4W, CdZU)

R4: Although training our model requires considerable computing resources as mentioned in Appendix Section A, the runtime cost and resources required for the inference stage are affordable. Our model can run on a single NVIDIA RTX 4090 GPU (24GB) thanks to our two-stage training/inference since one does not need to load all models simultaneously. The total inference time for one image composition can be finished in 20 seconds on an RTX 4090 (these include predicting the depth map, running the SAM model to remove the background of the reference image, using MLLM to predict the 2.5D location, depth fusion, and image composition, the DDIM sampler is set as 50 steps). We will release our code and pre-trained models to the public for the final version.

---

### Decision · Program_Chairs · 2024-09-25

**Decision:**

Accept (poster)

**Comment:**

This paper proposes to integrate depth information into instruction-based image composing. The authors also introduce a counterfactual dataset of image-text pairs for the task of object location prediction. Both qualitative and quantitative results demonstrate the effectiveness of the proposed approach. After a rebuttal discussion period, this paper received all positive reviews. The reviewers unanimously agreed that the idea is interesting, the dataset is novel, and the performance is good compared to the state-of-the-art methods. AC recommends that the authors incorporate all reviewers' feedback into the final camera-ready version and congratulates the paper on its acceptance.